# Magnetic Losses in Soft Ferrites

**Samuel Dobák** [1] , **Cinzia Beatrice** [2] , **Vasiliki Tsakaloudi** [3] **and Fausto Fiorillo** [2,*]

1 Institute of Physics, Faculty of Science, P. J. Šafárik University, 04154 Košice, Slovakia; samuel.dobak@upjs.sk
2 Advanced Materials Metrology and Life Science Division, Istituto Nazionale di Ricerca Metrologica-INRIM, 10135 Torino, Italy; c.beatrice@inrim.it
3 Center for Research and Technology Hellas CERTH, Thermi, 57001 Thessaloniki, Greece; vikaki@certh.gr
* Correspondence: f.fiorillo@inrim.it; Tel.: +39-011-3919841

**Abstract:** We review the basic phenomenology of magnetic losses from DC to 1 GHz in commercial and laboratory-prepared soft ferrites considering recent concepts regarding their physical interpretation. This is based, on the one hand, on the identification of the contributions to the magnetization process provided by spin rotations and domain walls and, on the other hand, the concept of loss separation. It additionally contemplates a distinction between the involved microscopic dissipation mechanisms: spin damping and eddy currents. Selected experimental results on the broadband behavior of complex permeability and losses in Mn-Zn ferrites provide significant examples of their dependence on sintering methods, solute elements, and working temperature. We also highlight the peculiar frequency and temperature response of Ni-Zn ferrites, which can be heavily affected by magnetic aftereffects. The physical modeling of the losses brings to light the role of the magnetic anisotropy and the way its magnitude distribution, affected by the internal demagnetizing fields, acts upon the magnetization process and its dependence on temperature and frequency. It is shown that the effective anisotropy governs the interplay of domain wall and rotational processes and their distinctive dissipation mechanisms, whose contributions are recognized in terms of different loss components.

**Keywords:** soft ferrites; magnetic losses; loss separation; magnetic resonance; permeability dispersion

## 1. Introduction

The sustained trend toward increasing operating frequencies and the miniaturization of components and devices in power electronics is faced with certain hurdles when dealing with the weight and size of the employed inductive cores. These are made, for the most part, of sintered soft ferrites, the material of choice for high-frequency inductors and inductive cores. Their widespread use for energy transfer in a variety of devices (e.g., Switch-Mode Power Supplies, converters, etc.) relates to a combination of solidly assessed and inexpensive preparation technology, compact core shape, and low energy losses. However, a reduced core size and increased operating frequencies may engender relevant thermal effects, easily favored by the low thermal conductivity of the material, with an ensuing drift of the magnetic response of the inductive components. Control and minimization of the energy losses are indeed the basic objectives of the research on improved ferrites, but any route towards increased efficiency passes through the physical understanding of the dissipative processes across a broad range of frequencies. The associated phenomenology is quite complex. It is the result of contributions by domain wall (dw) and rotational processes. They share the magnetization reversal in a frequency-dependent mode and release energy to the lattice either by conduction electrons (eddy currents) or by a degradation of the motion of the precessing spins, both inside the domains and the moving dws [1].

The favorite approach to optimal soft magnetic properties of ferrites is, besides preparation and sintering strategies, one of doping, where additional oxides are introduced in order to modify both the intrinsic and structural parameters and, eventually, minimize

the energy losses at the working temperature [2]. For example, by introducing oxides like $SiO_2$, CaO, and $Nb_2O_5$ in Mn-Zn ferrites (base composition $MnZnFe_2O_4$), we can change the nature and resistivity of the grain boundaries, whereas $TiO_2$, $SnO_2$, and CoO dissolve in the lattice and the corresponding cations enter the $Fe^{3+}$ sites, thereby modifying both the intragrain resistivity and magnetic anisotropy [3,4]. Ni-Zn ferrites (base composition $NiZnFe_2O_4$) are frequently enriched mainly for cost reasons by Cu additions, where $Cu^{2+}$ ions tend to substitute $Ni^{2+}$ ions in the octahedral sites [5]. Co doping is also exploited in order to improve the high-frequency permeability. Again, magnetic anisotropy is the main intrinsic parameter modified by the introduction of additional elements, besides possible interference by diffusing ions with the dw motion [6,7]. Both direct and indirect effects on the magnetic losses and their temperature dependence can thus be obtained with the help of foreign cations. Aside from the magnetic anisotropy, non-stoichiometric compositions altering the balance between $Fe^{2+}$ and $Fe^{3+}$ cations can also modify the electrical conductivity, with ensuing effects on the loss figure and its frequency dependence. This goes hand-in-hand with the behavior of the complex permeability $\mu = \mu' - j\mu''$, and whenever the (*J*–*H*) hysteresis cycle (*J* and *H* being magnetic polarization and field, respectively) can be reasonably described by an elliptical loop, the energy loss per unit volume versus frequency is obtained as

$$W(f) = \pi J_{\mathrm{p}}^2 \mu'' / \left(\mu'^2 + \mu''^2\right) = \pi H_{\mathrm{p}}^2 \mu'', \qquad \left[\mathrm{J/m^3}\right] \tag{1}$$

for a given peak polarization value $J_{\mathrm{p}}$ (or peak field value $H_{\mathrm{p}}$). The description of the hysteresis loop in terms of real $\mu'$ and imaginary $\mu''$ permeability components is an acceptable approximation up to medium $J_{\mathrm{p}}$ values (say, up to around $J_{\mathrm{p}} \sim J_{\mathrm{s}}/2$, if $J_{\mathrm{s}}$ is the saturation polarization) and is increasingly appropriate under increasing frequencies. However, the simple structure of Equation (1) conceals certain difficulties in assessing a coherent interpretation and prediction of the magnetic losses on a wide range of frequencies, ideally from a quasi-static response to a fully relaxed magnetization process at very high frequencies. This calls for a theoretical framework befitting the results such as those shown in Figures 1 and 2, where significant examples of broadband $\mu'(f)$ and $W(f)$ behaviors, measured up to 1 GHz in commercial ring samples of Mn-Zn and Ni-Zn ferrites, are provided. The symbols denote the results obtained for the defined $J_{\mathrm{p}}$ values of 2 mT and 10 mT by the fluxmetric technique [8]. The continuous lines are obtained by the measurement of $\mu'(f)$ and $\mu''(f)$ by a transmission line method (sweep mode by a Vector Network Analyzer, VNA) and calculation of $W(f)$ by Equation (1). The comparison between the two materials shows (Figure 1a) that an inverse relationship between the DC permeability value and the cutoff frequency is approximately satisfied in accordance with Snoek's rule. The non-monotonic frequency dependence of $W(f)$ in the Ni-Zn ferrites, in contrast with the behavior of the same quantity in the Mn-Zn samples (Figure 1b), points to a complex dw evolution with frequency in these materials and a quite different sharing of the dissipation mechanisms between dw displacements and rotations.

The physical interpretation and modeling of magnetic losses in soft ferrites across the extremely broad range of frequencies shown in Figure 1 call for a quantitative appraisal of the separate roles of dw and rotational processes and the related dissipation mechanisms, that is, eddy currents and spin damping. It is an objectively difficult task and empirical approaches are usually proposed in the literature. Loss calculations are in fact frequently performed by elaborating on the Steinmetz equation, modified and improved according to specific applicative needs [9–13]. The basic formulation is a simple power-law dependence on $J_{\mathrm{p}}$ and $f$ of the specific power loss

$$P(f) = k J_{\mathrm{p}}^{\beta} f^{\alpha}, \qquad \left[\mathrm{W/m^3}\right] \tag{2}$$

where the empirical parameters $k$, $\alpha$, and $\beta$ are adjusted and defined by identification and best fitting. This is possible, however, for a limited interval of frequency and $J_{\mathrm{p}}$ values only, and it does not contemplate the case of non-sinusoidal induction. Generalized versions of

the Steinmetz equation have therefore been proposed, an example of which is provided by the expression

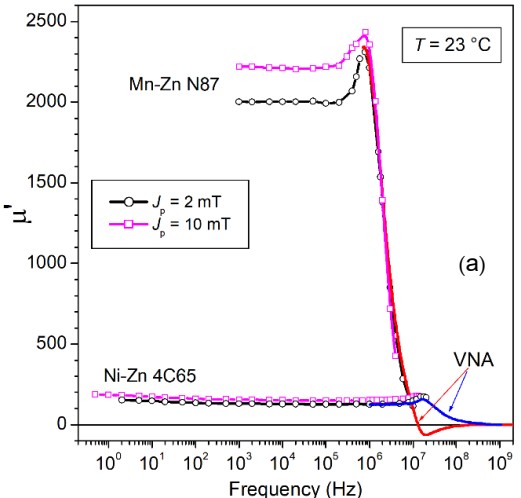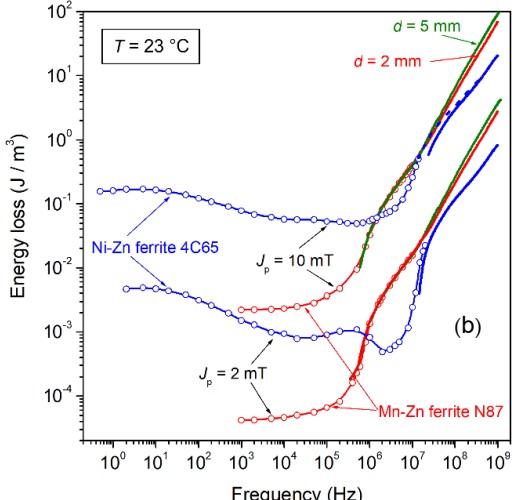

**Figure 1.** Real permeability component $\mu'(f)$ (**a**) and energy loss $W(f)$ (**b**), measured versus magnetizing frequency, up to 1 GHz, in commercial Mn-Zn and Ni-Zn ferrites. The symbols denote the results obtained, up to a few MHz, by fluxmetric measurements. The peak polarization values are $J_P = 2$ mT and $J_P = 10$ mT. The continuous lines covering the upper-frequency range are obtained by a transmission line method (Vector Network Analyzer, VNA). Equation (1) is employed to obtain the permeability from the fluxmetrically measured loss $W(f)$ at defined $J_P$ value. The same equation provides $W(f)$ from knowledge of the VNA-measured $\mu'(f)$ and $\mu''(f)$. The eddy current contribution to $W(f)$ is singled out beyond some 10 MHz by measurements on 5 mm and 2 mm thick Mn-Zn ring samples.

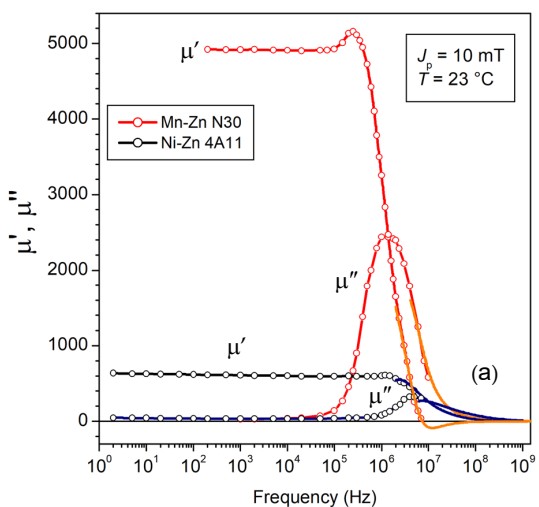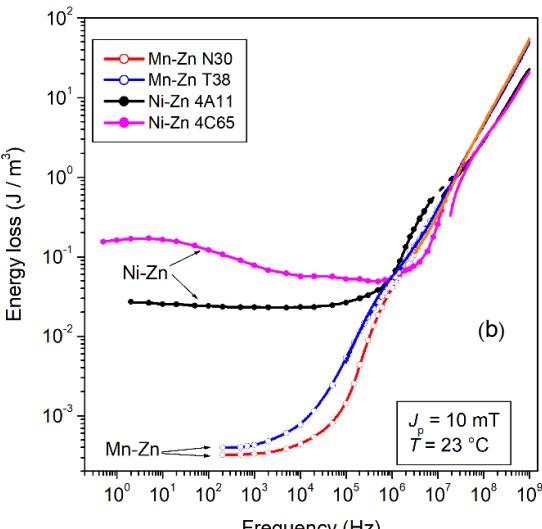

**Figure 2.** (**a**) Real $\mu'(f)$ and imaginary $\mu''(f)$ permeabilities measured up to 1 GHz at defined peak polarization $J_P = 10$ mT (symbols: fluxmetric measurements; continuous lines: VNA measurements) in two commercial types of Mn-Zn and Ni-Zn ferrites. Their behaviors roughly comply with Snoek's rule. (**b**) Corresponding broadband energy losses $W(f)$. The Ni-Zn ferrites, while exhibiting dramatically higher losses than Mn-Zn ferrites at low frequencies, eventually display similar and eventually lower $W(f)$ values upon attaining the MHz range. This points to quite different roles of dw and rotational processes in these two materials, ultimately descending from the largely different magnetic anisotropy constant.

$$P = \left(\frac{1}{T}\right) \int_0^T k_1 \left|\frac{\mathrm{d}B}{\mathrm{d}t}\right|^{\alpha} |B(t)|^{\beta-\alpha} \mathrm{d}t, \qquad \left[\mathrm{W/m^3}\right] \tag{3}$$

where $T$ is the magnetization period. Equation (3) is typically applied for linear piecewise (triangular, trapezoidal) induction waveforms below the MHz range [12,13]. Equivalent circuit modeling, relying on a linear $J$–$H$ DC relationship, is often invoked in order to emulate the dispersion of the complex permeability versus frequency and the related resonance effects using umped $R$, $L$, and $C$ parameters [14,15]. In fact, ferrite-cored inductors used in Switch-Mode Power Supplies are treated as a rule through their equivalent $L$–$R$ circuit, but, for practical reasons, the working point of the core can be moved toward the strongly nonlinear region around the knee of the magnetization curve [16]. Modeling of the inductor loss, in this case, relies on the description of the nonlinear equivalent $R$ and $L$ resistance and inductance by multiparameter functions and their identification for any defined working frequency by an appropriate set of measurements under a standard sinusoidal flux [17].

It is apparent that these models have value in terms of fitting procedures but lack any direct connection with the phenomenology of the magnetization process, and do not permit any conclusion regarding the relationship between the losses and the intrinsic and structural properties of the material and the poor feedback toward improved preparation methods. A step forward towards a physical appraisal is one of considering $W(f)$ as the sum of a quasi-static term $W_{\mathrm{h}} = \lim_{f \to 0} W(f)$ (hysteresis loss) and a dynamic one $W_{\mathrm{dyn}}(f)$. Such decomposition is not only suggested by the $W(f)$ behavior (Figure 1b), but it is the obvious conclusion of the physical analysis of the magnetization process [18]. A practical way to unfold this analysis consists of expressing the magnetic field at any given point of a generic dynamic $J(H)$ loop as $H = H_{\mathrm{h}}(J) + H_{\mathrm{dyn}}(J, \mathrm{d}J/\mathrm{d}t)$ [19,20]. With the function $H_{\mathrm{h}}(J)$ obtained for any $J_{\mathrm{p}}$ value, either by experiment or modeling, and $H_{\mathrm{dyn}}(J, \mathrm{d}J/\mathrm{d}t)$ found by the measurement of an adequate sequence of hysteresis loops traced under constant $\mathrm{d}J/\mathrm{d}t$, a surface in the $(H, J, \mathrm{d}J/\mathrm{d}t)$ space is identified. The hysteresis loops and losses are then reconstructed for different (e.g., sinusoidal) exciting regimes in the very same material.

A true physical interpretation of the magnetic losses in ferrites requires insight into the nature of the magnetization process and its evolution with frequency. We identify this process with the combination of dw motion and rotation of the spins inside the domains. In fact, because of the low value taken by the anisotropy constant, from some 10–100 $\mathrm{J/m^3}$ in Mn-Zn up to a few $\mathrm{kJ/m^3}$ in Ni-Zn [21,22] in the range 20–120 °C, rotations can remarkably contribute to the magnetization reversal. They have reversible characteristics and the associated susceptibility $\chi_{\mathrm{rot}}$ is at maximum for vanishing applied fields. It is observed, in particular, that the Rayleigh law is generally followed up to a good fraction of the normal magnetization curve (say around 50–100 mT if $J_{\mathrm{s}}$ is of the order of 400–500 mT). One can then describe to good approximation the magnetic susceptibility as

$$\chi(H) = a + bH = (\chi_{\mathrm{rot}} + \chi_{\mathrm{w,rev}}) + \chi_{\mathrm{w,irr}}(H), \tag{4}$$

where $a$ and $b$ are the Rayleigh constants and the dw susceptibility is composed of a reversible $\chi_{\mathrm{w,rev}}$ and an irreversible $\chi_{\mathrm{w,irr}}(H) = bH$ part. The rotations do not participate in the hysteresis (quasi-static) energy loss, which is expressed as $W_{\mathrm{h}} = \frac{4}{3}\chi_{\mathrm{w,irr}}\mu_0 H^2$, but they will share with the dws the main dynamic dissipative phenomena, spin damping, and eddy currents. There is a scale difference, however, that permits one to identify from a physical viewpoint the different contributions to the magnetic losses and provide a solid quantitative base to the concept of loss decomposition. It is a concept whose predicting properties are well-established for magnetic steels across the range of frequencies important for applications [18,23]. In ferrites, however, the phenomenology is more complex, and the frequency range covered under a non-negligible magnetic response by the material can go from quasi-static to a few GHz. Equations (1) and (4) usefully suggest that, within the stated approximation of low-to-moderate $J_{\mathrm{p}}$ values, we can regard permeability and losses

in a unified fashion, from both an experimental and a theoretical viewpoint. This we will discuss and clarify in the following sections, together with a survey of recent results on commercial and laboratory Mn-Zn and Ni-Zn samples.

## 2. Basic Phenomenology and Theoretical Approach to Energy Losses

We have provided in Figure 1a a significant example of broadband loss and permeability behavior in commercial Mn-Zn and Ni-Zn sintered ferrites. A further example is given in Figure 2, where we showed the results of measurements performed on standard materials at a defined peak polarization value $J_P$ = 10 mT. The broadband response of these ferrites is achieved by a combination of fluxmetric and transmission line measurements. The fluxmetric setup is centered on a hysteresisgraph-wattmeter, by which one can determine energy loss, hysteresis loop, and complex permeability on ring samples, up to a maximum frequency of 20 MHz. These measurements partly overlap in frequency with the measurements made up to about 1 GHz by means of a transmission line setup centered on the use of a VNA in reflection mode. The complex permeability is obtained, starting from a few hundred kHz, with the ring sample placed against the bottom of the shorted line. The energy loss is then calculated through Equation (1). A full discussion of the summarized measuring methods and setups was provided in previous works [8,23]. We can nevertheless observe in Figure 1 that the fluxmetric (symbols) and VNA (lines) results merge upon a more or less extended interval at high frequencies, although the former are obtained at a defined $J_P$ level and the latter under a defined exciting power (typically around a few mW). This occurs because the dw displacements increasingly give way to spin rotations, an intrinsically linear process, under increasing frequencies. Consequently, at higher $J_P$ values there is a reduction in the overlapping frequency interval (see Figure 1b). Distinguishing between dw and rotational contributions, the magnetization process is indeed central to the assessment of permeability and loss behaviors versus $f$ and $J_P$. We need, however, to identify the associated microscopic dissipation channels. To start with, we might single out the role of eddy currents and the related release of energy to the lattice by the conduction electrons. Such a role can be neglected in the Ni-Zn ferrites, whose resistivity values are generally higher than $10^5$ $\Omega$m. This is not the case with the Mn-Zn ferrites, where, as shown in Figure 3, the DC resistivity is around a few $\Omega$m and decreases by more than an order of magnitude upon attaining the MHz range. The reason for such a decrease is recognized in the heterogeneous structure of the Mn-Zn sintered ferrites, where the individual semiconducting grains are separated by a few nanometer-thick insulating boundaries. Therefore, the electrical measurements versus temperature evidence a behavior synthesized by the equivalent RC scheme shown in Figure 3a, with the real $\rho'(f)$ and imaginary $\rho''(f)$ components following the classical Debye relaxation equations

$$\rho'(f) = \rho_g + \frac{\rho_b}{1 + \omega^2\tau^2}\rho''(f) = \rho_b\,\omega\tau/\left(1 + \omega^2\tau^2\right) \tag{5}$$

where $\rho_g$ and $\rho_b$ are the resistivities of the grain and grain boundary, respectively, and the time constant $\tau = \rho_b\varepsilon'$, with $\varepsilon'$ the real DC permittivity of the grain boundary. The example provided in Figure 3 concerning the case of a 4000 ppm CoO-doped Mn-Zn ferrite, shows that the experimental $\rho'(f)$ and $\rho''(f)$ behaviors conform to the relaxation-dispersion Equation (5), with $\rho'(f)$ reducing to $\rho_g$ beyond a few MHz. It is noted in Figure 3 how the DC resistivity (i.e., the resistivity of the grain boundaries $\rho_b$) is decreased by a factor of around 6 on passing from room temperature to 100 °C. The ohmic conductivity of the near-insulating boundaries (estimated around $10^{-3}$–$10^{-5}$ $\Omega^{-1}$ m$^{-1}$ [24]) actually appears to decrease with the temperature $T$ according to the activation law $\sigma = \sigma_0 exp\left(-\frac{E_a}{k_B T}\right)$, with the activation energy $E_a$ estimated for the ferrites of Figure 3 around 0.21 eV.

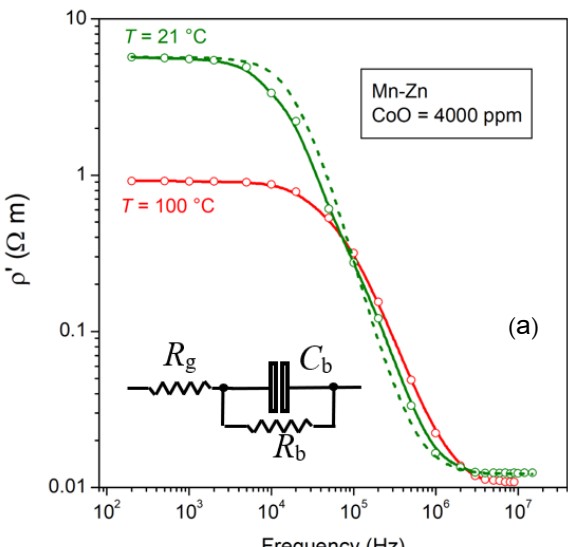
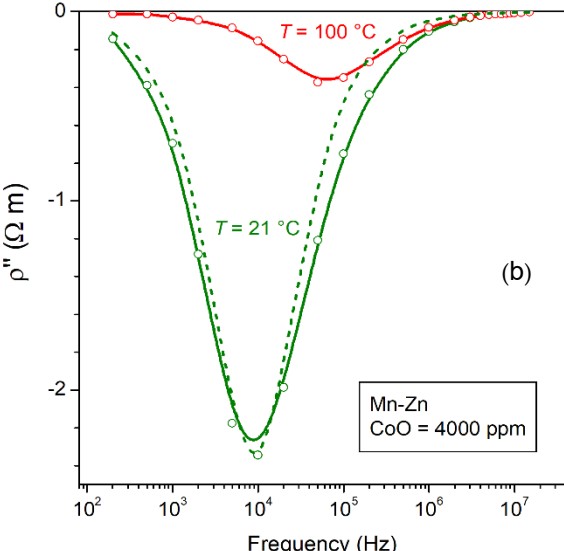

**Figure 3.** An example of real ρ′(f) (**a**) and imaginary (**b**) ρ″(f) resistivity behaviors, as obtained by measurements with the four-point method up to 10 MHz on a CoO-doped Mn-Zn sintered ferrite. The observed dependences on frequency comply with an equivalent RC circuit, where $C_b$ and $R_b$ lump the capacity and resistance of the grain boundaries and $R_g \ll R_b$ denotes the intragrain resistance. The dashed lines are predicted by the relaxation Equation (5). To note the limiting lower value attained by ρ′(f) in the MHz range where it reduces to the intragrain resistivity $ρ_g$. (Partly adapted from Ref. [24]).

### 2.1. Eddy Current Losses

The non-negligible overall conductivity of the Mn-Zn ferrites naturally calls for an evaluation of the eddy current losses and their distinctive contribution to the behavior of the measured losses. The conventional approach to such an evaluation is centered, in general, on the formulation of the total energy loss as the sum of separate components, although no consensus exists regarding the precise nature and role of such components. A popular scheme is one of defining the energy loss at a given frequency as

$$W(f) = W_h + W_{eddy}(f) + W_r(f) \tag{6}$$

where the frequency-independent quasi-static (hysteresis) component $W_h$ combines with the dynamic eddy-current $W_{eddy}(f)$ and the residual $W_r(f)$ components, the latter incorporating all possible extra contributions to energy dissipation [25,26]. $W_{eddy}(f)$ is usually assumed to pertain to a homogeneous material and as such, it is calculated [26–30]. $W_r(f)$ is generally associated with the damping of the precessional spin motion and the related resonance phenomena, but a role by dielectric losses is sometimes invoked [26,31], besides the possible extra eddy-current effects by quantum charge tunneling through the grain boundary layers at high frequencies [32]. However, the matter of quantitatively describing the magnetic loss and its broadband frequency dependence in soft ferrites calls for a proper physical context, where the identification of the relevant dissipation mechanisms is associated, in a real material, with their statistical properties. This is the approach worked out in metallic soft magnetic materials through the Statistical Loss Model [18] and later extended to the behavior of the poorly conducting and heterogeneous Mn-Zn ferrites [33]. To start with, we recognize in the hysteresis term $W_h$ an exclusive contribution by the irreversible dw displacements (Barkhausen jumps), which occur under quasi-static excitation. These jumps combine to provide the macroscopic magnetization reversal in a time-independent fashion. The related energy dissipation is due to the extremely localized eddy currents generated by the microscopic dw jumps (time constant around $10^{-9}$–$10^{-10}$ s) in combination with the frictional torque Γ affecting the simultaneous fast precession of the spins

inside the wall as lumped in the Landau-Lifshitz phenomenological constant $\alpha_{LL}$. The hysteresis loss is generally modeled by formal approaches (e.g., the Preisach model), given the complexity of the magnetization process and its description in terms of first principles. It is generally accepted to look at it as a sort of generalized coercivity $H_c$, defined according to the relationship $W_h(J_p) = 4 H_c J_{p,dw}$, where it is assumed that the peak polarization $J_p$ is attained by the joint contributions of dw displacements and rotations $J_p = J_{p,dw} + J_{p,rot}$. Low anisotropy (from some 10 J/m$^3$ in Mn-Zn to some kJ/m$^3$ at most in Ni-Zn) is a peculiar property of soft ferrites, engendering the relevant role of the rotational process. For sufficiently small $J_p$ values, we can write $J_{p,dw} = J_p \cdot \mu_{dwDC}/(\mu_{dwDC} + \mu_{rotDC})$, where the components $\mu_{dw,DC}$ and $\mu_{rot,DC}$ of the DC permeability $\mu_{DC}$ can be experimentally singled out as discussed below. On the other hand, the coercive field reflects the fluctuating profile of the dw energy landscape, influenced by the lattice defects and the average grain size $<s>$, and we can write $H_c = a(J_p)\gamma_w/(J_s\langle s \rangle)$, where $\gamma_w$ is the dw energy, $J_s$ is the saturation polarization, and $a(J_p)$ is an increasing function of $J_p$, resulting from the distribution of the pinning field strength. Since $\gamma_w = 2\sqrt{AK}$, where A~$10^{-11}$ J/m is the stiffness constant and $K$ is the anisotropy constant, we arrive at expressing the hysteresis loss as

$$W_h(J_p) \sim 4H_c J_{p,dw} = 8a(J_p) \cdot \frac{\sqrt{AK}}{J_s\langle s \rangle} J_p \mu_{dwDC}/(\mu_{dwDC} + \mu_{rotDC}) \qquad (7)$$

We notice in Figure 2 the dramatic increase in $W_h$ on passing from the Mn-Zn to the Ni-Zn ferrites and how this correlates with a corresponding decrease in the permeability. This is consistent with Equation (7), because, as previously remarked, $K$ is much larger and the grain size $<s>$ much lower (about an order of magnitude) in Ni-Zn. It is apparent in Figure 2b that the somewhat anomalous low-frequency behavior of $W(f)$ occurs in the Ferroxcube 4C65-type NiZn ferrite. As discussed later, this is an effect of dw relaxation, associated with a process of atomic/electronic diffusion, endowed with an average activation energy $E_a$ = 0.66 eV. We can classify it according to the conventional rule stated in Equation (6) as a dynamic term lumped in $W_r(f)$. However, before entering the discussion on the physical meaning and properties of $W_r(f)$, we need to clarify the role of the eddy currents and how one can derive $W_{eddy}(f)$ in terms of dw and rotational processes.

It is actually established and physically assessed that $W_{eddy}(f)$ can be derived as the sum of the classical $W_{cl,eddy}(f)$ and excess $W_{exc,eddy}(f)$ terms [18]. The latter descends from the discrete nature of the magnetization process by dws and the eddy currents circulating around the moving walls, whereas $W_{cl}(f)$ is calculated by making an abstraction of the dws [34]. Mn-Zn ferrites cannot actually be treated like the conventional electrical steel sheets given their peculiar conductivity mechanism and their frequency dependent electrical properties. As suggested by the example shown in Figure 3 and demonstrated by Finite Element Method analysis [24], the eddy current patterns are generally confined within the insulating grain boundaries at low frequencies to eventually encompass the whole cross-sectional area of the sample on attaining the frequencies at which the grain boundary capacitive reactance tends to vanish. Given its local, grain-delimited character, $W_{exc,eddy}(f)$ relates to the intragrain conductivity $\sigma_g$, which is of the order of 20 $\Omega^{-1}$ m$^{-1}$ (compared to 2 × 10$^6$ $\Omega^{-1}$ m$^{-1}$ in Fe-3wt%Si). According to the statistical loss model [34], we can calculate the maximum value attainable by $W_{exc,eddy}(f)$ at any frequency by considering the ideal limiting situation where at any instant of time the whole irreversible process is conveyed by a single dw. We obtain in such a case, as shown in [24], that for a sample of cross-sectional area S

$$W^{(max)}_{exc,eddy}(f) = 2\pi^2 \sigma_g GSf J^2_{p,dw}, \qquad \left[\text{J/m}^3\right] \qquad (8)$$

where the adimensional constant $G$ = 0.1356 and $J_{p,dw} = J_p\, \mu_{dw}/\mu$ is the fractional dw contribution to $J_p$. It turns out that the quantity $W^{(max)}_{exc,eddy}(f)$ is far lower than the measured energy loss at all frequencies [24]. A dynamic loss component generated by the dw motion

is nevertheless observed [35]. It is generated by a frictional effect on the precessing spins, consistent with the contribution $J_{\mathrm{p,dw}}(f)$.

In evaluating the eddy current losses, we are therefore left with the analysis of the macroscopic patterns of the induced currents (the so-called eddy current classical loss $W_{\mathrm{cl,eddy}}(f)$), whose role will expectedly increase under increasing frequencies. This effect is already apparent in Figure 1, where we observe a slight increase in $W(f)$ in the commercial Mn-Zn ferrite EPCOS N87, beyond about 10 MHz, upon the increase in the sample thickness from 2 mm to 5 mm. Another example is shown in Figure 4a, where $W_{\mathrm{cl,eddy}}(f)$ can be appreciated at frequencies higher than 6 MHz in 5.02 mm thick ring samples. We conclude that, in general, eddy current losses do play a role at very high frequencies only. Figure 4b refers, however, to the quite limiting case of the commercial ferrite EPCOS T38, which is characterized by a low DC resistivity ($8.5 \times 10^{-2}$ $\Omega$m versus 6.3 $\Omega$m of the EPCOS N87 type ferrite). With large and thick samples (in this case, rings of 9.49 mm $\times$ 5.25 mm cross-sectional area) of this type, $W_{\mathrm{cl,eddy}}(f)$ already starts to contribute to $W(f)$ at around $10^4$ Hz. The theoretical approach to $W_{\mathrm{cl,eddy}}(f)$ developed in Ref. [24] makes it clear that in this case, the grain boundaries have relatively low resistivity (about two orders of magnitude lower than in the N87-type ferrites), and the dissipation by macroscopic eddy-current paths already matters in the kHz range.

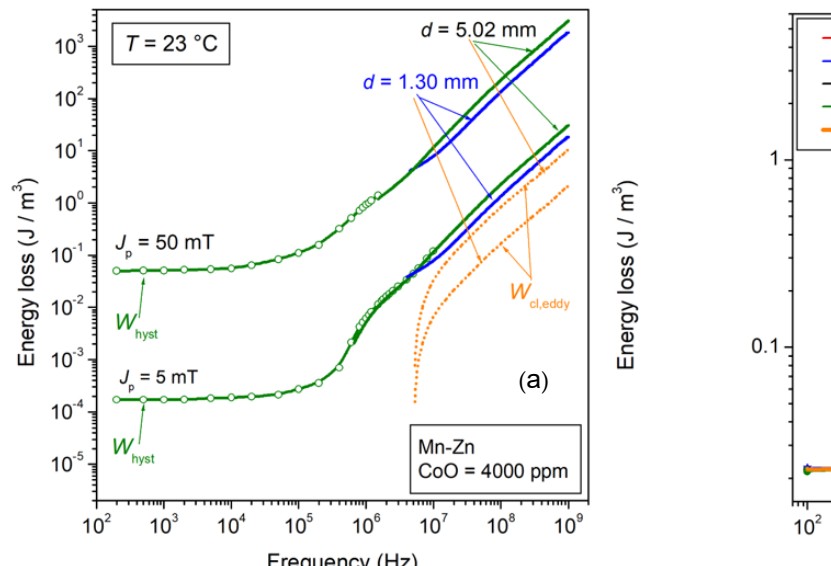
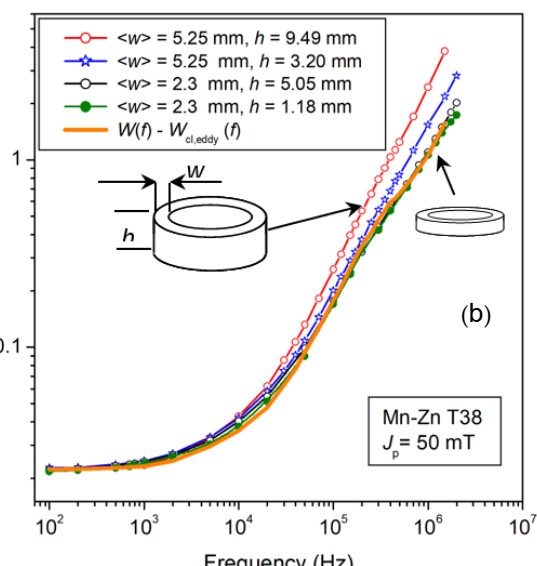

**Figure 4.** (**a**) Energy loss $W(f)$ versus frequency in a CoO-doped Mn-Zn ferrite ring and its dependence on the sample thickness $d$. With $d$ reduced from 5.02 mm to 1.30 mm, $W(f)$ is affected, beyond a few MHz, by the decrease in the eddy current losses $W_{\mathrm{cl,eddy}}(f)$, which attain vanishing values. Such a decrease is predicted by means of a simplified Equation (10) (dashed lines). (**b**) Very large and thick low-resistivity T38 Mn-Zn ring samples can be affected by the eddy current losses already in the kHz range. In such a case, $W_{\mathrm{cl,eddy}}(f)$ is calculated through Equation (9) and, once subtracted from the measured $W(f)$, the eddyfree energy loss curve $W_{\mathrm{h}} + W_{\mathrm{r}}(f)$ (solid line) is obtained. (Partly adapted from Ref. [24]).

The problem of predicting $W_{\mathrm{cl,eddy}}(f)$ in the electrically heterogeneous structure of the Mn-Zn ferrites is treated in [24] by solving Maxwell's equations via a multiscale approach, which is applied to a regular array of square semiconducting grains separated by a thin semi-insulating layer. The following Boltzmann-type closed expression is consequently worked out

$$W_{\mathrm{cl,eddy}}(J_{\mathrm{P}},\, f) = \left[ \frac{(k_0 - k_\infty)\mathrm{A}}{1 + exp(m\, log_{10}(f/f_0))} + k_\infty A \right] J_{\mathrm{p}}^2 f, \quad [\mathrm{J/m^3}] \qquad (9)$$

where $k_0 A J_p^2$ and $k_\infty A J_p^2$ are the limiting values attained by $W_{\text{cl,eddy}}(J_p, f)/f$ for $f \to 0$ and $f \to \infty$, respectively [24]. The quantity $A$ is an effective area, depending on the actual cross-sectional area of the sample modulated by the width-to-thickness ratio; $m$ and $f_0$ are fitting parameters. The basic property of Equation (9) is one of describing the transition between the low-frequency and high-frequency regimes, as embodied by the relaxation behavior of the measured resistivity. A somewhat blurred transition may occur in the previous T38 sample. It turns out, in any case, that, by subtracting $W_{\text{cl,eddy}}(J_p, f)$ from the experimental $W(f)$ found in rings of different sizes and thicknesses, we arrive at the thickness-independent eddy current free loss curve illustrated in Figure 4b. Under many circumstances, however, (see, for example, Figures 1 and 4a) eddy currents start to play a role at so high a frequency that the difference between intragrain and intergrain currents has already disappeared and the standard formulation for $W_{\text{cl,eddy}}(J_p, f)$ in a material of conductivity $\sigma' = \rho'/(\rho'^2 + \rho''^2)$ and rectangular cross-sectional area S of aspect ratio R is applied

$$W_{\text{cl,eddy}}(J_p, f) = \left(\frac{\pi^2}{6}\right)\sigma'(f)\cdot(12k(R)S)J_p^2 f, \qquad \left[\text{J/m}^3\right] \tag{10}$$

where $\sigma'(f)$ is the real part of the measured conductivity and $k(R)$ is an increasing function of $R$, given by $k(R) = 0.0744R - 0.0434R^2$ [36]. It saturates, for $R = 1$, at the value $k(R) = 0.035$.

The dashed lines in Figure 4a, calculated by Equation (10), can then account for the high-frequency thickness-related bifurcation of $W(f)$. For a sample of a circular cross-section of diameter d, we equivalently have

$$W_{\text{cl,eddy}}(J_p, f) = \left(\frac{\pi^2}{16}\right)\sigma'(f)d^2 J_p^2 f \qquad \left[\text{J/m}^3\right] \tag{11}$$

We can estimate the skin effect at the highest frequencies (hundreds of MHz) by taking into account that the dw processes have fully relaxed and the spin damping has brought the intrinsic permeability (the one independent of eddy currents) $|\mu|$ close to $\mu_0$. The calculations show that at the highest frequency of 1 GHz the skin depth is of the order of 2 mm, and Equations (10) and (11) acceptably estimate $W_{\text{cl,eddy}}(f)$. Note that the frequently invoked dielectric losses are basically lumped, if any, in the calculation of Maxwell's equations leading to Equation (9). On the other hand, they are not expected to play a role because the overall volume of the dielectric regions is negligible with respect to the sample volume.

*2.2. Spin Damping, DW Motion, Rotations, and Broadband Permeability*

We have verified that the role of the eddy current losses $W_{\text{eddy}}(f)$ in Mn-Zn ferrites, as introduced in Equation (6), can be theoretically predicted and singled out in terms of classical loss $W_{\text{cl,eddy}}(f)$, the one pertaining to the macroscopic eddy current patterns. We are thus left in Equation (6), besides the hysteresis component $W_h$, with the so-called residual dynamic term $W_r(f)$. This reflects the frictional effects affecting the spin precession inside the moving dws and the magnetic domains (spin damping, sd). We shall appropriately talk, in this case, of loss decomposition into a dynamic dw component $W_{\text{exc,sd}}(J_p, f)$ and a rotational component $W_{\text{rot,sd}}(J_p, f)$. The near insulating properties of the ferrites favor then, with the possible exception of very large and thick samples (like the T38 Mn-Zn ferrites in Figure 4b) suitably treated for very high DC permeability by excess $Fe^{2+}$ ions, the direct release of the magnetic energy to the lattice via spin-orbit coupling in place of the scattering by conduction electrons [37]. The experiments show, in particular, that the motion of a domain wall traveling at speed $\dot{x}$ in a non-conducting ferrite single crystal obeys the equation

$$2J_s(H - H_c) = \beta_{\text{sd}}\dot{x} \tag{12}$$

once the coercive field $H_c$ is overcome by the pressure provided by the applied field $H$ [35]. The damping coefficient $\beta_{\text{sd}}$ is related to the Landau-Lifshitz damping coefficient $\alpha_{\text{LL}}$

according to the equation $\beta_{\text{sd}} = \alpha_{\text{LL}}(2J_s/\mu_0\gamma\delta)$, where $\gamma$ is the electron gyromagnetic ratio and $\delta$ is the dw thickness [38]. The same Equation (12) with $\beta_{\text{sd}}$ substituted by the eddy current damping coefficient $\beta_{\text{eddy}} = 4\sigma G J_s^2 d$, is used to describe the motion of a dw in a slab of thickness d and provides the starting formulation in the development of the Statistical Theory of Losses [18]. We are therefore in a position to draw on this theory in the assessment of the dynamic loss $W_{\text{exc,sd}}(J_p, f)$, which compounds with $W_h$ to provide the dw contribution $W_{\text{dw}}(f)$ to $W(f)$. This quantity, however, goes hand in hand with the complex permeability, and a consistent interpretative approach calls for the joint appraisal of $\mu = \mu' - j\mu''$ and $W(f)$ and their decomposition into dw and rotational contributions. Having already identified $W_h(J_p)$ and evaluated the contribution $W_{\text{eddy}}(J_p, f) \cong W_{\text{cl,eddy}}(J_p, f)$, we focus on the dynamic contribution $W_r(J_p, f)$, the so-called residual term, which, as put in evidence in the $W(f)$ behaviors shown in Figures 1, 2 and 4, is far from residual in most cases. We write then Equation (1) by assuming $\mu'' = \mu''_{\text{dw}} + \mu''_{\text{rot}}$

$$W_r(J_p, f) = W_{\text{exc}}(J_p, f) + W_{\text{rot}}(J_p, f) = \pi J_p^2 (\mu''_{\text{dw}} + \mu''_{\text{rot}}) / \left(\mu'^2 + \mu''^2\right), \qquad \left[\text{J/m}^3\right] \qquad (13)$$

where with the usual notation $W_{\text{exc}}(J_p, f)$ we identify the dw contribution to the dynamic loss. Basically, we aim at separating the dw and rotational contributions to $W_r(J_p, f)$, which implies a corresponding separation of the associated permeabilities. Once this feat is accomplished by a direct analysis of the experimental quantities, we can proceed toward the theoretical assessment of the whole phenomenology. We shall first analyze the experimental $\mu'(f)$ and $\mu''(f)$ behaviors in order to achieve, at least within the Rayleigh region, the decomposition

$$\mu'(f) = \mu'_{\text{dw}}(f) + \mu'_{\text{rot}}(f) \qquad \mu''(f) = \mu''_{\text{dw}}(f) + \mu''_{\text{rot}}(f) \qquad (14)$$

We point towards an estimate of the quantities involved in Equation (14) by taking advantage of the following assumptions regarding dw displacements and rotations [39]. (1) The orientations of the easy axes are uniformly distributed (all quadrants included) and the associated rotational permeability

$$\mu_{\text{rot,DC}} = 1 + J_s^2 / 3\mu_0 K \qquad (15)$$

where the anisotropy constant $K$ is typically of the order of 20–50 J/m$^3$, is constant with good approximation within the Rayleigh region [39]. For the materials evoked in Figures 1–4, this is verified up to $J_p \sim 50$ mT (that is $J_p/J_s \sim 0.1$). (2) The DC imaginary permeability is exclusively associated with the dw displacements $\mu''_{\text{DC}} \equiv \mu''_{\text{dw,DC}}$. In fact, rotations are not involved with the hysteresis loss. (3) At least two values $J_{p1}$ and $J_{p2}$ such that $\mu''_{\text{dw,DC}}(J_{p1}) \gg \mu''_{\text{dw,DC}}(J_{p2})$ are experimentally available within the above-mentioned $J_p$ range. Following the Rayleigh law, where the irreversible magnetization is provided by the term $bH^2$ [18], we have

$$\mu'' = (4b/3\pi)H_p \qquad \mu''(J_{p1})/\mu''(J_{p2}) = H_p(J_{p1})/H_p(J_{p2}) \qquad (16)$$

The permeability decomposition shown in Figure 5 is carried out for $J_{p1} = 50$ mT and $J_{p2} = 2$ mT and the experimental magnetization curve (see inset in Figure 5a) shows that $H_p(50 \text{ mT}) \sim 23 \cdot H_p(2 \text{ mT})$, that is $\mu''_{\text{dw,DC}}(50 \text{ mT}) \gg \mu''_{\text{dw,DC}}(2 \text{ mT})$. Following then the condition (1), extended to the AC regime, we write

$$\mu'(J_{p1}, f) - \mu'(J_{p2}, f) = \mu'_{\text{dw}}(J_{p1}, f) - \mu'_{\text{dw}}(J_{p2}, f) \qquad (17)$$

$$\mu''(J_{p1}, f) - \mu''(J_{p2}, f) = \mu''_{\text{dw}}(J_{p1}, f) - \mu''_{\text{dw}}(J_{p2}, f) \qquad (18)$$

and from condition (3) we have

$$\mu''(J_p = 50 \text{ mT}, f) - \mu''(J_p = 2 \text{ mT}, f) \approx \mu''_{\text{dw}}(J_p = 50 \text{ mT}, f). \qquad (19)$$

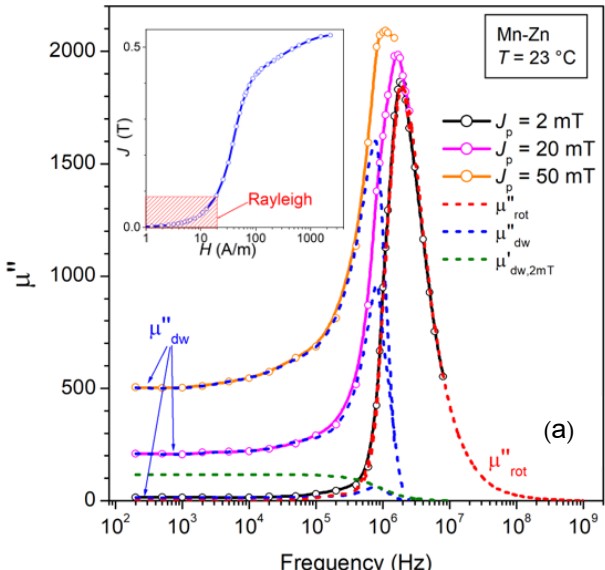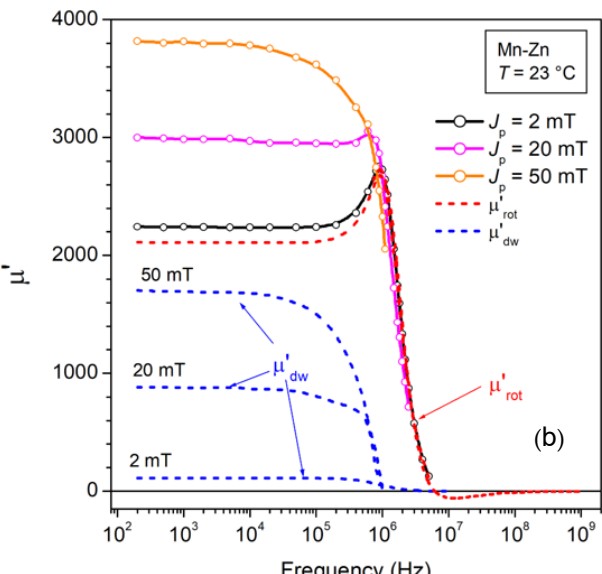

**Figure 5.** Imaginary (**a**) and real (**b**) permeabilities are decomposed into their dw and rotational contributions. The adopted procedure, carried out for $J_P$ values belonging to the Rayleigh region (see inset in Figure 5a), is illustrated and justified through Equations (14)–(24). $\mu'_{dw}$ and $\mu''_{dw}$ relax around 1–2 MHz, whereas $\mu'_{rot}$ and $\mu''_{rot}$ display behavior consistent with the unfolding of ferromagnetic resonance.

The so-obtained $\mu''_{dw}(J_P = 50\,\text{mT}, f)$ is normalized to the DC value, as imposed by condition (2) ($\mu''_{dw,DC}(J_P = 50\,\text{mT}) = \mu''_{DC}(J_P = 50\,\text{mT})$), and we can approximately evaluate the rotational ($J_P$ independent) imaginary permeability as

$$\mu''_{rot}(f) \approx \mu''(J_P = 50\,\text{mT}, f) - \mu''_{dw}(J_P = 50\,\text{mT}, f). \tag{20}$$

$\mu''_{rot}(f)$ is a function starting from zero value for $f = 0$, as required by the condition of zero hysteresis loss by rotations. We consequently obtain the approximate value of $\mu''_{dw}(J_P = 2\,\text{mT}, f)$ as

$$\mu''_{dw}(J_P = 2\,\text{mT}, f) \approx \mu''(J_P = 2\,\text{mT}, f) - \mu''_{rot}(f). \tag{21}$$

The dashed lines in Figure 5a show the frequency behaviors of the calculated $\mu''_{dw}(J_P = 2\,\text{mT}, f)$, $\mu''_{dw}(J_P = 50\,\text{mT}, f)$, $\mu''_{dw}(J_P = 20\,\text{mT}, f)$, and $\mu''_{rot}(f)$. We note how $\mu''_{rot}(f)$ becomes practically coincident with the experimentally measured $\mu''(J_P = 2\,\text{mT}, f)$ on entering the MHz range. The same figure shows the estimate of the real dw component $\mu'_{dw}(J_P = 2\,\text{mT}, f)$, which is calculated from $\mu''_{dw}(J_P = 2\,\text{mT}, f)$ on the assumption of a Debye's relationship

$$\mu'_{\text{dw}}\left(J_{\text{P}} = 2\,\text{mT}, f\right) \approx \mu'_{\text{dw,DC}}\left(J_{\text{P}} = 2\,\text{mT}\right) \cdot \frac{1}{1 + (f/f_1)^2} \tag{22}$$

where, at the relaxation frequency $f_1$ (in this case around 800 kHz), the imaginary permeability attains the maximum value $\mu''_{\text{dw}}(J_{\text{P}} = 2\,\text{mT}, f_1) = (1/2)\,\mu'_{\text{dw,DC}}(J_{\text{P}} = 2\,\text{mT})$.

The rotational real component is immediately obtained as

$$\mu'_{\text{rot}}(f) \approx \mu'\left(J_{\text{P}} = 2\,\text{mT}, f\right) - \mu'_{\text{dw}}\left(J_{\text{P}} = 2\,\text{mT}, f\right). \tag{23}$$

Figure 5b shows that the initial permeability is largely of rotational origin, while being affected by the landmark features (e.g., the hump in the $\mu'(f)$ behavior) of the ferromagnetic resonance. It is also apparent that the uncertainty due to the approximation of Equation (21) on the calculated $\mu'_{\text{dw}}(J_{\text{P}} = 2\,\text{mT}, f)$ curve has little effect on the obtained behavior of $\mu'_{\text{rot}}(f)$. The permeability $\mu'_{\text{dw}}(J_{\text{P}}, f)$, calculated up to $J_{\text{P}} = 50\,\text{mT}$ as

$$\mu'_{\text{dw}}\left(J_{\text{P}}, f\right) \approx \mu'\left(J_{\text{P}}, f\right) - \mu'_{\text{rot}}(f) \tag{24}$$

is shown to fully relax around 1 MHz, consistent with the observed coalescence of the $\mu'(J_{\text{P}}, f)$ curves beyond such a frequency. Earlier dispersion and relaxation of the dw processes with respect to rotations is a well-known effect in metallic soft magnetic materials such as amorphous and nanocrystalline tapes [40]. The moving dws reverse locally the magnetization, in contrast with the far smoother variation in space and time of the magnetization simultaneously occurring by spin rotations inside the domains. They are then subjected to higher eddy current counterfields and more effective damping. Similar conclusions can be drawn for dissipation by spin damping. We can express, in particular, the frictional torque $\boldsymbol{\Gamma}$ affecting the dynamics of the magnetic moments inside the moving walls through the Gilbert equation

$$|\boldsymbol{\Gamma}| = (\alpha_{\text{LL}}/\gamma\mu_0 J_s)\left|J \times \frac{\text{d}J}{\text{d}t}\right| = (\alpha_{\text{LL}}/\gamma\mu_0)J_s\frac{\text{d}\theta}{\text{d}t} \tag{25}$$

where $\alpha_{\text{LL}}$ is the Landau-Lifshitz damping constant, $\gamma$ is electron gyromagnetic ratio, and $\text{d}\theta/\text{d}t$ is the angular velocity of the precessing spins [38]. A much larger $\boldsymbol{\Gamma}$ is evidently associated with the motion of the dws with respect to rotations. The dw motion will not be sustained by the applied field at the highest frequencies, as illustrated by the example shown in Figure 5.

### 2.3. The Energy Loss by Spin Damping and Its Decomposition

The previous results and the discussion concerning the behaviors of $\mu'(f)$, $\mu''(f)$, their dw and rotational components, and the pre-emptive discrimination of the eddy current losses by Equations (9)–(11), lead us to interpret the residual loss $W_{\text{r}}(f)$ in terms of spin damping. To this end, we shall take profit of the direct relationship existing between complex permeability and loss according to Equation (13), and we will describe the viscous rotational response of the spins inside the magnetic domains to the applied AC field by the Landau–Lifshitz–Gilbert (LLG) equation. For a single domain endowed with saturation magnetization $M_s$ and anisotropy field $H_k$ normally directed with respect to the exciting field of frequency $\omega = 2\pi f$, the response of the spin assembly is given, in terms of the real and imaginary relative permeabilities, by the equation [38]

$$\mu'_{\text{rot,sd}}(\omega, H_k, \pi/2) = 1 + \chi'_{\text{rot,sd}}(\omega, H_k, \pi/2) = 1 + \frac{\omega_H\omega_M\left[\omega_H^2 - (1 - \alpha_{\text{LL}}^2)\omega^2\right]}{\left[\omega_H^2 - (1 + \alpha_{\text{LL}}^2)\omega^2\right]^2 + 4\alpha_{\text{LL}}^2\omega_H^2\omega^2} \tag{26}$$

$$\mu''_{\text{rot,sd}}(\omega, H_k, \pi/2) = \chi''_{\text{rot,sd}}(\omega, H_k, \pi/2) = \frac{\alpha_{\text{LL}}\omega_M\omega\left[\omega_H^2 + (1 + \alpha_{\text{LL}}^2)\omega^2\right]}{\left[\omega_H^2 - (1 + \alpha_{\text{LL}}^2)\omega^2\right]^2 + 4\alpha_{\text{LL}}^2\omega_H^2\omega^2} \tag{27}$$

where $\omega_H = \gamma\mu_0 H_K$ and $\omega_M = \gamma\mu_0 M_s$. Since we are dealing with a polycrystalline material, besides the angular distribution of the easy axes, we need to take into account the

relevant role of the intergrain demagnetizing fields, which can significantly interfere and combine with the relatively small magnetocrystalline anisotropy fields. This eventually results in an amplitude distribution of an effective anisotropy field $H_{k,eff} = 2K_{eff}/J_s$, which contributes, together with damping, to the broadening of the magnetic resonance peaks in Figure 5. This is equivalent to saying that resonant absorption and dissipation of energy by the precessing spins will occur across a broad range of frequencies. We describe this distribution by means of the lognormal function

$$g(H_{k,eff}) = (1/\sqrt{2\pi}\sigma H_{k,eff}) \, \exp\left[-\frac{(\ln(H_{k,eff}) - h)^2}{2\sigma^2}\right] \tag{28}$$

where $h = \langle\ln(H_{k,eff})\rangle$ and $\sigma$ is the standard deviation of $\ln(H_{k,eff})$ [39]. We assume an isotropic distribution of the effective easy axes through the function $p(\theta) = sin\theta$, with $0 \leq \theta \leq \pi/2$ and we integrate over amplitude and orientation of the effective anisotropy field, assuming that $g(H_{k,eff})$ and $p(\theta)$ are independent functions. After averaging the angular distribution of the easy axes, we obtain the susceptibilities

$$\langle\chi'_{rot,sd}(f)\rangle = \frac{2}{3}\int_0^\infty g(H_{k,eff})\chi'_{rot}\left(f, H_{k,eff}, \frac{\pi}{2}\right) dH_{k,eff} \tag{29a}$$

$$\langle\chi''_{rot,sd}(f)\rangle = \frac{2}{3}\int_0^\infty g(H_{k,eff})\chi''_{rot}\left(f, H_{k,eff}, \frac{\pi}{2}\right) dH_{k,eff} \tag{29b}$$

We actually wish to provide a formulation possibly covering the whole broad range of frequencies (DC-1 GHz) available to the experiments. It is apparent, looking at the permeability behaviors shown in Figures 1 and 5, that this means, in general, to deal with the magnetic response of the ferrites till it becomes vanishingly small. In fact, as the grains increasingly pass through magnetic resonance under increasing frequencies, they become transparent to the AC field and unable to compensate for the magnetic poles appearing on the still-active surrounding grains, whose resonance frequency will inevitably increase. A grain merged in a medium of susceptibility $|\chi(f)|$ will be endowed with an effective demagnetizing coefficient $N_{d,eff} \sim N_d/(1 + |\chi(f)|)$, so that, with decreasing $|\chi(f)|$ at high frequencies (see Figure 5), the local demagnetizing fields are expected to increase by a quantity

$$\Delta H_d \propto \Delta N_{d,eff} \approx \left(\frac{N_d}{1 + \chi_{DC}} - \frac{N_d}{1 + |\chi(f)|}\right) \cong N_d\frac{1 - |\chi(f)|/\chi_{DC}}{1 + |\chi(f)|} \tag{30}$$

assuming $\chi_{DC} \gg 1$ [39]. Consequently, the distribution $g(H_{k,eff})$ will broaden towards increasingly higher $H_{k,eff}$ values under increasing frequency, following the decrease in the total susceptibility $|\chi(f)|$. This effect is lumped into a frequency-dependent quantity $h$ in Equation (27) according to

$$h(f) = \left\langle\ln\left(H_{k,eff}^{(DC)} + C\frac{1 - \frac{|\chi(f)|}{\chi_{DC}}}{1 + |\chi(f)|}\right)\right\rangle, \tag{31}$$

with $C$ a suitable constant. The permeabilities $\mu'_{rot,sd}(f)$ and $\mu''_{rot,sd}(f)$ predicted by Equations (29a) and (29b) (dashed lines) are compared with the experimental components $\mu'_{rot}(f)$ and $\mu''_{rot}(f)$ in the example provided in Figure 6a. The rotational loss $W_{rot,sd}(J_p, f)$ correspondingly calculated by means of Equation (13) is equally shown by the dashed lines in Figure 6b for two $J_p$ values. $W_{rot,sd}(J_p, f)$ fits the high-frequency behavior of the experimental $W(J_p, f)$ in thinned samples, where the eddy current losses are negligibly small (these having already been calculated through Equations (9)–(11)). An overall view of the DC-1 GHz experimental $W(J_p, f)$ behavior, obtained for the $J_p$ values spanning two orders of magnitude, is provided in Figure 7a. We notice that the high-frequency losses are fully accounted for by the calculated $W_{rot,sd}(J_p, f)$. This occurs, in particular, starting from about 800 kHz at $J_p = 2$ mT and 5 MHz for $J_p = 200$ mT. The role of dws is obviously enhanced with increasing $J_p$. $W_h(J_p)$ follows in fact a $\sim J_p^{2.8}$ law, versus the

$J_p{}^2$ behavior exhibited by $W_{\text{rot,sd}}(J_p, f)$, but earlier relaxation versus frequency of the dw processes will inevitably result in a merging of $W(J_p, f)$ into $W_{\text{rot,sd}}(J_p, f)$. Figure 7b shows an example of loss decomposition, whereby subtracting the hysteresis loss $W_h(J_p)$ and the theoretically predicted $W_{\text{rot,sd}}(J_p, f)$ from the measured $W(J_p, f)$, one can single out the dynamic excess loss component $W_{\text{exc,sd}}(J_p, f)$. This quantity directly descends from the frictional effects on the travelling domain walls as described by Equation (12) and lumped in the damping coefficient $\beta_{\text{sd}}$. It is observed to follow a power law dependence on frequency $W_{\text{exc,sd}}(J_p, f) \propto f^n$, with $n{\sim}0.9$ until, on approaching the MHz range, the dw motion is fully superseded by the rotations and the related losses ($W_h(J_p)$) and $W_{\text{exc,sd}}(J_p, f)$ disappear. It is remarkable that the same Equation (12), where the spin damping coefficient $\beta_{\text{sd}}$ is substituted by the eddy current coefficient $\beta_{\text{eddy}}$, is the starting point for the excess loss prediction in magnetic sheets, according to the statistical theory of losses [18]. Such a theory was actually generalized to spin damping dissipation [33], one main difference residing in a power-law coefficient $n$ not far from 1 for $W_{\text{exc,sd}}(J_p, f) \propto f^n$. The value $n = 0.5$ is generally found in soft magnetic sheets [34]. It is realized that this difference originates in the lack of eddy current counter fields in ferrites. This makes the evolution of the statistics of the dw processes, which is at the core of the prediction of the excess loss in metallic alloys, weakly dependent on frequency. The theory offers the tools for dealing with such a scenario and the related quantitative prediction, as discussed in Ref. [41]. It leads to the expression

$$W_{\text{exc,sd}}(J_p, f) \cong 4.4 J_{p,\text{dw}} \left( \beta_{\text{sd}} S V_0^m J_{p,\text{dw}} f / \langle s \rangle J_s^2 \right)^{\frac{1}{1+m}} \quad (m \leq 1), \ [\text{J/m}^3] \quad (32)$$

where $J_{p,\text{dw}} < J_p$ accounts for the fractional contribution to $J_p$ by the dw motion and $V_0$ is a statistical parameter related to the distribution of the dw pinning fields. A more general expression for of $W_{\text{exc,sd}}(J_p, f)$, approximated by Equation (32), is also available [42]. For the case shown in Figure 7b, $m{\sim}0.1$ and the increase in $W_{\text{exc,sd}}(J_p, f)$ with $f$ is quasi-linear. The downfall of $W_{\text{exc,sd}}(J_p, f)$ (associated with that of $W_h(J_p)$) on entering the MHz range is apparent in Figure 7b.

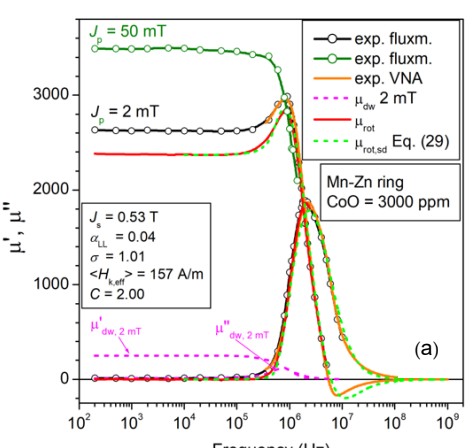 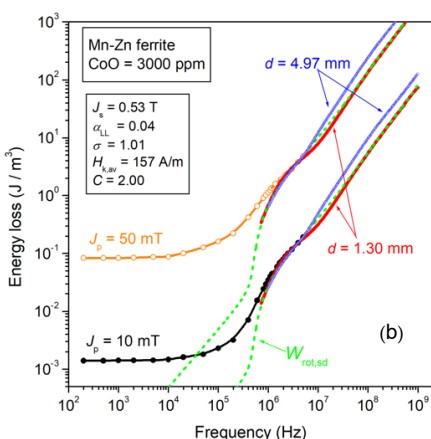

**Figure 6.** (**a**) The rotational permeability components $\mu'_{\text{rot}}(f)$ and $\mu''_{\text{rot}}(f)$ are singled out in a CoO-doped Mn-Zn ferrite ring sample, following the scheme provided by Equations (19)–(23). The symbols and the solid lines are the results of fluxmetric (exp. fluxm.) and transmission line (exp. VNA) measurements. They are theoretically calculated using Equations (29a) and (29b) (dashed lines) with lognormal distribution function for the effective anisotropy field strength $H_{k,\text{eff}}$ (average value $<H_{k,\text{eff}}>$). The parameters of the distribution, given in the inset, are the width $\sigma$ and the constant $C$ (Equations (27) and (31)). $\alpha_{\text{LL}}$ is the Landau–Lifshitz damping constant. (**b**) The corresponding rotational loss by spin damping $W_{\text{rot,sd}}(f)$ is obtained from the calculated rotational permeabilities by means of Equation (13). The additional contribution by eddy currents is observed beyond a few MHz in the thicker sample ($d = 4.97$ mm).

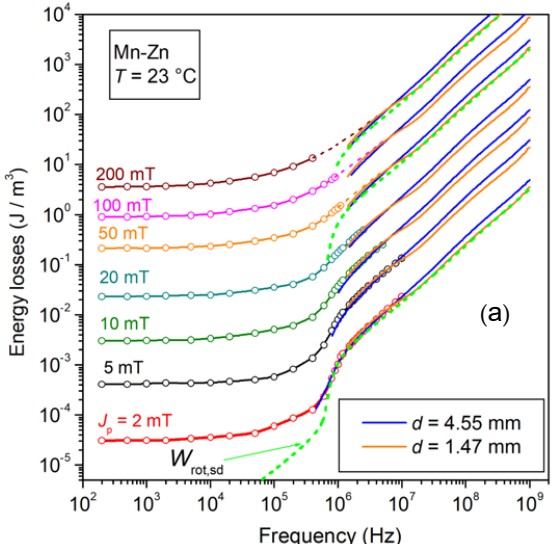
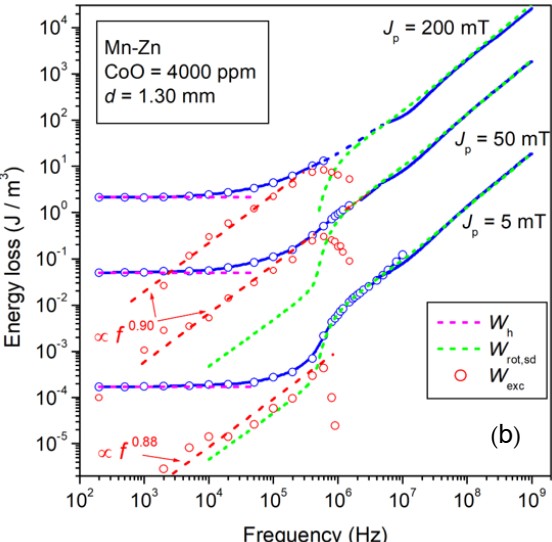

**Figure 7.** (**a**) Broadband energy loss in an Mn-Zn ferrite ring measured from $J_p = 2$ mT to $J_p = 200$ mT before and after reduction of the sample thickness $d$ from 4.55 mm to 1.47 mm. An eddy current loss contribution is observed before thinning in the upper-frequency range ($f > 10$ MHz). The rotational loss by spin damping $W_{rot,sd}(f)$, calculated by Equations (29a) and (29b) (dashed lines), fully predicts the measured loss in the thinner ring from 800 kHz and 3 MHz onward for $J_p = 2$ mT and $J_p = 200$ mT, respectively. (**b**) $W(J_p, f)$ is decomposed as $W(J_p, f) = W_h(J_p) + W_r(J_p, f) = W_h(J_p) + W_{exc,sd}(J_p, f) + W_{rot,sd}(J_p, f)$. The excess loss $W_{exc,sd}(J_p, f)$ is described by Equation (32) and drops on approaching the MHz range, together with $W_h(J_p)$, where the dws come to full rest.

## 3. A Few Applicative Examples of Mn-Zn Ferrites

We summarize in this section a few significant results regarding the role of material treatment and of measuring temperature on the broadband energy losses of Mn-Zn ferrites. We will provide an assessment of the loss phenomenology based on the concepts and methods developed in the previous sections. The role of the effective anisotropy energy, the one ensuing from the combined action of the anisotropy field and the internal demagnetizing fields, is fundamental in balancing the competing effects of dw displacements and spin rotations. Composition, preparation method, microstructure, and temperature, all affect such a balance, whose understanding is a basic aim of the presently provided results and their analysis.

### 3.1. Energy Loss versus Sintering Temperature

Different sintering schedules affect the material microstructure, and for a defined composition we look for the optimal process leading to the best permeability and loss response. We consider here an example, where a ferrite made by mixing 70 wt% $Fe_2O_3$, 24 wt% MnO, and 6 wt% ZnO, with the addition of $NbO_2$ and CaO dopants, is subjected to sintering for times $t_{sinter}$ ranging between 3 h and 7 h in the temperature interval $1325\,°C \leq T_{sinter} \leq 1360\,°C$. Table 1 provides a list of the physical parameters of the selected products. It is observed that longer sintering times and higher temperatures bring about increased conductivity and larger grain size. Non-homogeneous grain growth and increased porosity occur at the same time so that the soft magnetic properties correspondingly deteriorate [39]. This is observed in Figure 8a,b, representing selected normal magnetization curves and the temperature dependence of the energy loss at 100 kHz and $J_p = 100$ mT. It is shown in Figure 8b that, on passing from the 3 h at 1325 °C to the 7 h at 1360 °C sintering process, the loss suffers an approximately 40 % increase, while always attaining a minimum value at 90°C. More specifically, we see in Figure 9a that the decrease in the calculated rotational permeability $\mu'_{rot,DC}$ upon sintering at 1360 °C can be interpreted, with the help of Equation (15), in terms of an increase in the effective anisotropy energy

$<K_{eff}>$ from ~50 J/m$^3$ to ~67 J/m$^3$. This associates, as shown in the same figure, with a decrease in $\mu'_{dw,DC}$ by virtue of the corresponding increase in the dw energy and the more defective structure of the material. The ensuing effects on the $W(f)$ behavior, shown in Figure 9b, consist of the obvious increase in the dw-related loss $W_{dw}(f) = W_h + W_{exc,sd}(f)$ upon $T_{sinter} = 1360$ °C and in its retarded drop on approaching the MHz range. The latter phenomenon is not only due to the increased $W_{dw}(f)$ at low frequencies, but, more subtly, to a slight drift of the distribution function $g(H_{k,eff})$ (Equation (27)) toward higher $H_{k,eff}$ values. This engenders an upward shift in the spectrum of the resonance frequencies and the related absorption of energy. We will come across conspicuous evidence of this effect in the forthcoming sections.

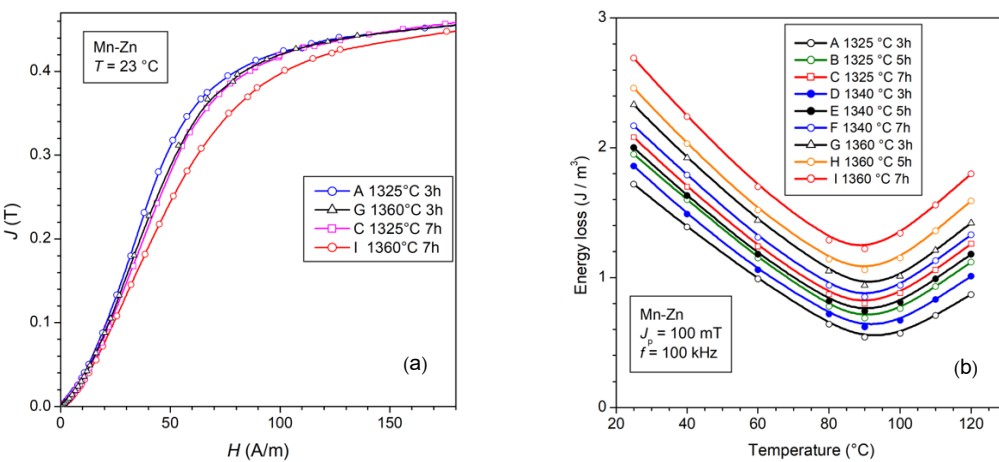

**Figure 8.** (**a**) Normal magnetization curve of samples sintered at 1325 °C and 1360 °C for different times. (**b**) The energy loss measured at 100 kHz attains minimum value at a temperature of 90 °C upon any sintering circumstance. Treatment at higher temperatures and longer times leads to increased porosity and structural inhomogeneities, aside from a moderate increase in the magnetic anisotropy, all leading to increased losses. (Adapted from Ref. [39]).

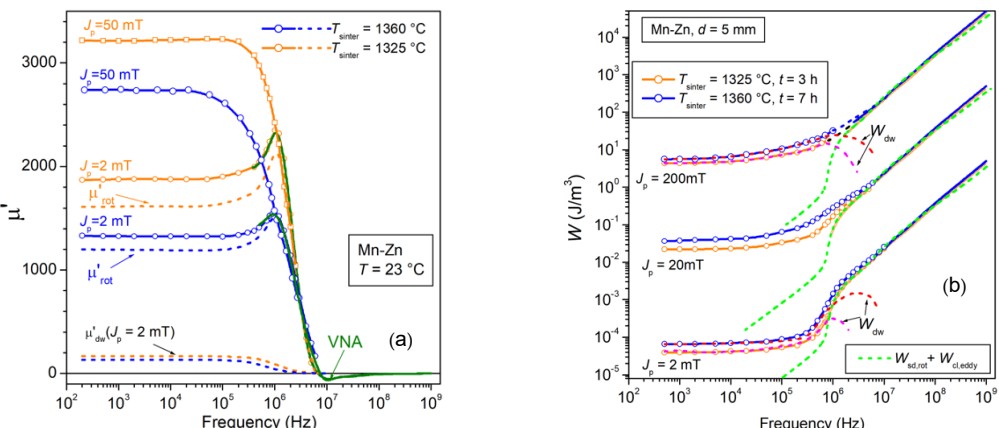

**Figure 9.** (**a**) The decrease in the permeability of the Mn-Zn ring samples occurring upon the increase in the sintering temperature $T_{sinter}$ from 1325 °C to 1360 °C is shared by the rotational $\mu'_{rot}$ and dw $\mu'_{dw}$ components. By interpreting Equation (15) in terms of effective magnetic anisotropy, the decrease in $\mu'_{rot,DC}$ descends from a variation of $<K_{eff}>$ from 50 J/m$^3$ to 67 J/m$^3$ following the increase of $T_{sinter}$ to 1360 °C. (**b**)The additional structural inhomogeneities brought about by higher $T_{sinter}$ lead to higher dw energy and stronger pinning centers. The dw contribution $W_{dw}(f) = W_h + W_{exc,sd}(f)$ to the measured loss $W(f)$ is correspondingly enhanced and survives up to higher frequencies, as concluded after subtracting the predicted high-frequency component $W_{rot,sd}(f) + W_{cl,eddy}(f)$ from $W(f)$.

**Table 1.** Physical parameters of the selected Mn-Zn ferrite ring samples. $T_s \equiv$ sintering temperature; $t_s \equiv$ sintering time; $\delta \equiv$ mass density; $<s> \equiv$ average grain size; $\sigma_{DC} \equiv$ DC conductivity; $\sigma_{10MHz} \equiv$ conductivity at 10 MHz; $J_s \equiv$ saturation polarization.

| Sample | $T_s$ (°C) | $t_s$ (h) | $\delta$ (kg/m³) | $<s>$ (μm) | $\sigma_{DC}$ ($\Omega^{-1}$ m$^{-1}$) | $\sigma_{10\,MHz}$ ($\Omega^{-1}$ m$^{-1}$) | $J_s$ (T) |
|---|---|---|---|---|---|---|---|
| A | 1325 | 3 | 4830 | 11.4 | 0.146 | 56.8 | 0.55 |
| D | 1340 | 3 | 4850 | 12.3 | 0.214 | 82 | 0.55 |
| G | 1360 | 3 | 4900 | 15.1 | 0.220 | 80 | 0.55 |
| I | 1360 | 7 | 4890 | 19.8 | 0.215 | 71 | 0.55 |

*3.2. Effect of the Working Temperature and the Role of CoO Doping*

The soft magnetic properties of the Mn-Zn ferrites can be flexibly modified either by regulating their stoichiometry or by introducing additives because in this way we can influence the grain growth and microstructure, affect the magnetic anisotropy through the mechanism of anisotropy compensation, and increase the resistivity of the grains and the grain boundaries by solute cations and segregated second phases [2,3,43,44]. For example, an excess of positive anisotropy $Fe^{2+}$ ions in combination with the negative anisotropy of the host lattice, permits one to attain a minimum loss value at temperatures $T$ = 80–100 °C, the typical working temperatures of the inductive components in power electronics. The case reported in Figure 8b shows how relevant the variation of the energy loss with temperature can be. Many applications require, however, a certain independence of the magnetic properties from the variation in the working temperature, a feature competitively displayed, for example, by the nanocrystalline alloys [45].

It is recognized that, by introducing suitable amounts of $Co^{2+}$ cations, typically via mixing CoO with the pre-fired powders, one can mitigate the loss and permeability dependence on the temperature of standard Mn-Zn ferrites [46–48]. These are generally also enriched by conventional additives such as CaO and $Nb_2O_5$. The involved physical mechanism is assumed to be one of temperature compensation of the negative anisotropy of the host $Fe^{3+}$ ions by the positive anisotropy of the $Co^{2+}$ ions, both occupying the same octahedral sites. The question is therefore posed regarding the possibility of estimating the anisotropy energy and its dependence on temperature and the way this dependence is reflected in the loss properties. This should occur through the role exerted by the anisotropy on the dw and rotational processes. We will, therefore, account for the effective anisotropy constant $<K_{eff}>$, the quantity combining the magnetocrystalline anisotropy, and the internal (i.e., grain-to-grain) magnetostatic effects. We observe in Figure 10 how the dependence of initial permeability and energy loss on the temperature $T$, measured between $-20$ °C and 130 °C, evolves with the CoO content. The latter is made to range between 0 and 6000 ppm (in weight). The most effective constraint on the variation in these quantities with $T$ is attained with CoO = 3000–4000 ppm. This is understood in terms of a correspondingly constrained variation of $<K_{eff}>$ versus $T$. We can verify this assumption by estimating $\mu'_{rot,DC} \cong \mu_{rot,DC}$ through the procedure described in Section 2.2 (Equations (20) and (23)) and through Equation (15). Figure 11 provides examples of the experimental evaluation of $\mu'_{rot}(f)$ and the corresponding prediction by Equation (29) of $\mu'_{rot,sd}(f)$. The matrix of values $<K_{eff}>$ $(T, CoO)$ over the whole investigated temperature and CoO ranges is presented in Figure 12, where the $<K_{eff}>$ vs. $T$ and CoO content curves exhibit coherent behaviors with respect to the initial permeability and the energy loss dependence on $T$. There is an obvious direct correspondence between $<K_{eff}>$ and the initial permeability because this quantity is largely due to $\mu'_{rot}(f) \sim \propto 1/<K_{eff}>$ (Equation (15), while $\mu'_{dw}(f) \sim \propto 1/<K_{eff}>^{1/2}$. For what concerns the energy loss, we need to separately consider the role of $W_h$, $W_{exc,sd}(f)$, and $W_{rot,sd}(f)$. To start with, we refer to the previously postulated Equation (7) for the hysteresis loss and apply it to the experimental $W_h(T, CoO)$ behavior. We show in Figure 13 two examples of $W_h$ (CoO) behaviors and their predictions at different temperatures and

different $J_p$ values. The dashed lines are calculated using Equation (7), where we have introduced the effective anisotropy constant $<K_{eff}>$ shown in Figure 12b, the $J_s(T)$ values obtained by Vibrating Sample Magnetometer measurements, the micrographically observed average grain size $<s>$ ranging between 10.8 and 13.1 μm, and the quantities $\mu_{rot,DC}$, $\mu_{dw,DC}(J_p) \cong \mu'_{dw,DC}(J_p)$, as previously described. The fitting parameter $a(J_p)$, independent of CoO content, increases with $J_p$ according to $a(J_p) \propto J_p{}^{1.1}$ and is approximately constant between 23 °C and 130 °C. A description of the preparation procedure of the materials and a complete list of the main structural, electrical, and magnetic parameters are provided in Ref. [48].

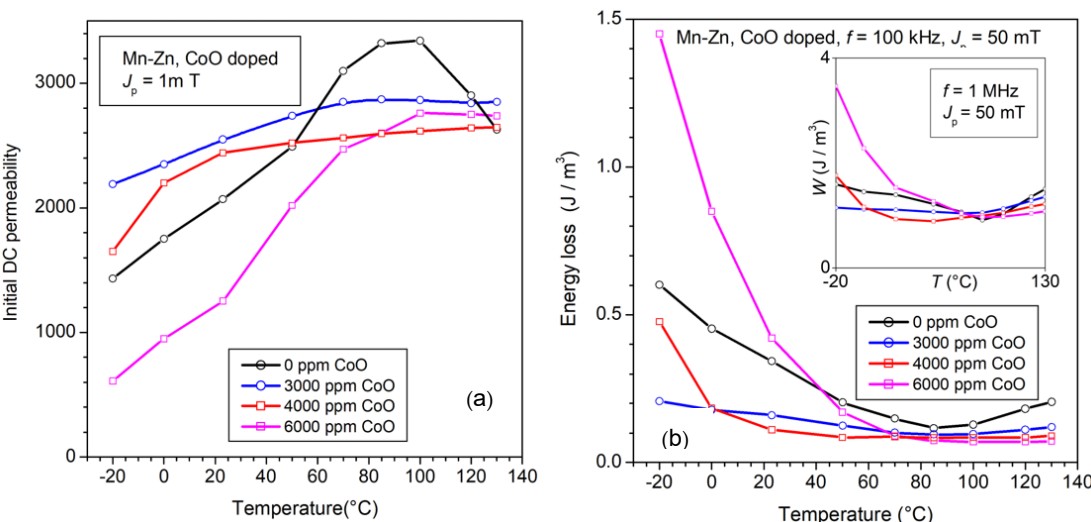

**Figure 10.** The dependence on the temperature *T* of the magnetic properties of Mn-Zn ferrites is mitigated by optimal CoO doping. We observe a smoothed behavior versus *T* of the DC initial permeability (**a**) and of the energy loss measured at 100 kHz and 1 MHz for $J_p = 50$ mT (**b**) for CoO addition of 3000–4000 ppm. (Adapted from Ref. [42]).

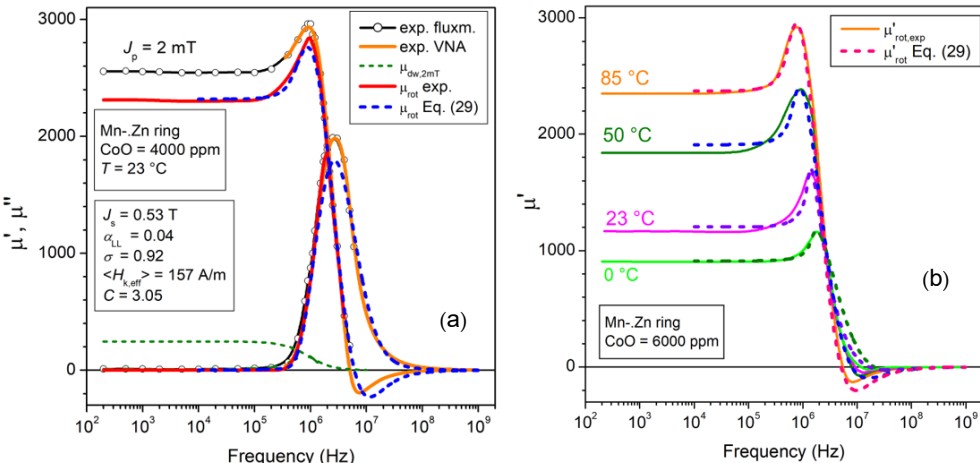

**Figure 11.** (**a**) Example of permeability decomposition in a 4000 ppm CoO-doped Mn-Zn ring sample. $\mu'_{rot}(f)$ and $\mu''_{rot}(f)$ are singled out according to Equations (20) and (23) (continuous lines) and compared with their theoretical prediction $\mu'_{rot,sd}(f)$ and $\mu''_{rot,sd}(f)$ (dashed lines), as derived from the Landau–Lifshitz equation and the assumed lognormal distribution for the effective anisotropy field $H_{k,eff}$ (Equation (29)). (**b**) Experimental $\mu'_{rot}(f)$ at different temperatures in a 6000 ppm CoO-doped sample and its prediction as $\mu'_{rot,sd}(f)$.

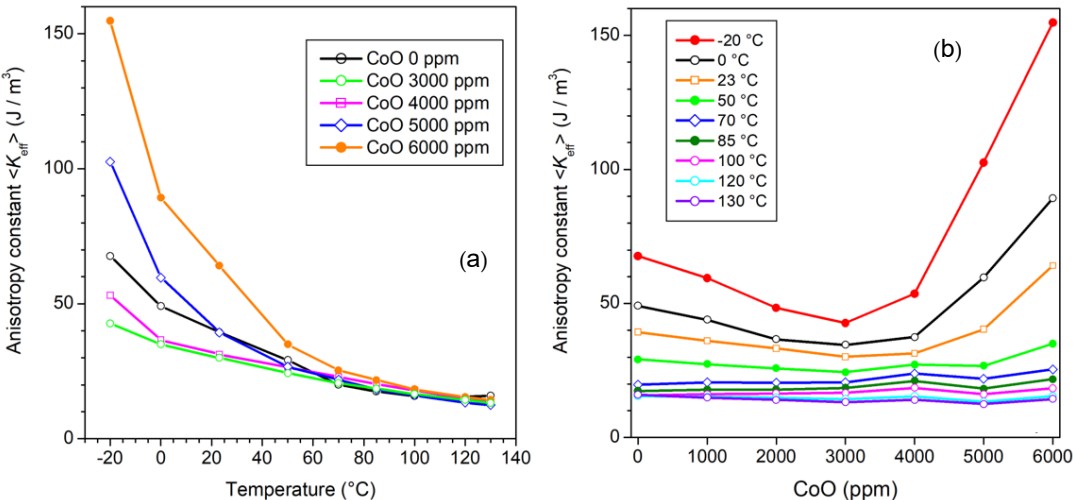

**Figure 12.** The effective anisotropy $<K_{eff}>$, calculated from knowledge of $\mu_{rot,DC}$, exhibits the slowest variation with temperature (**a**) after addition of 3000–4000 ppm CoO (**b**). (Adapted from Ref. [42]).

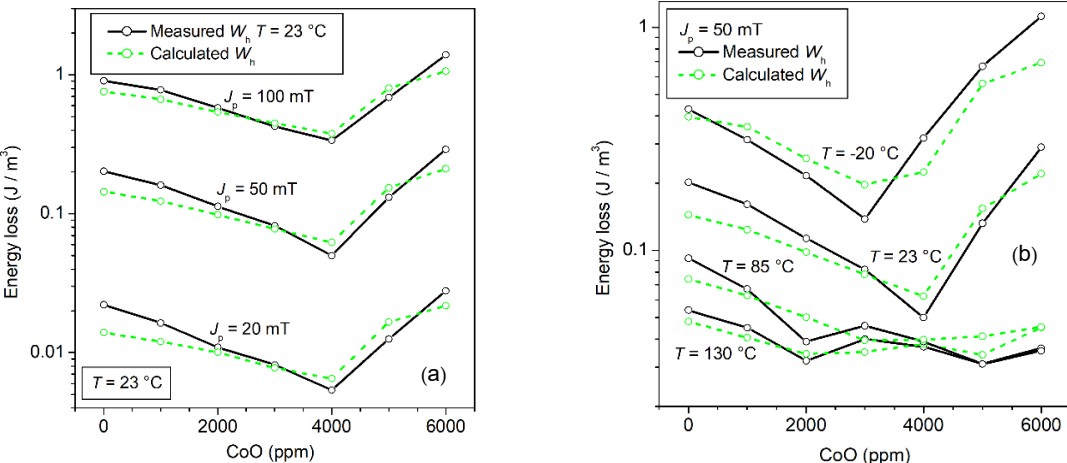

**Figure 13.** (**a**) The hysteresis energy loss $W_h$ measured at $T = 23\ °C$ versus CoO content at different $J_p$ values is compared with the prediction made through Equation (7). We make use in this equation of the experimental quantities $<K_{eff}>$ (Figure 12b), $J_s$, $\mu_{rot,DC}$, $\mu_{dw,DC}$, and $<s>$, and a parameter $a(J_p)$ $\sim\propto J_p^{1.1}$, independent of the CoO content. (**b**) The same predicting equation leads to the calculated $W_h$ versus CoO at defined $J_p$ value and different temperatures, broadly reflecting the evolution of $<K_{eff}>$ in Figure 12b at such temperatures. (Adapted from Ref. [42]).

The experimental loss decomposition and its theoretical interpretation are carried out once the properties of $W_h$ are at least qualitatively understood, starting with the determination of the rotational permeability components $\mu'_{rot}(f)$ and $\mu''_{rot}(f)$. This can be done according to the procedure described in Section 2.2, leading to results such as the ones illustrated in Figure 14a. It is shown here, in particular, that $\mu'_{rot}(f)$ and $\mu''_{rot}(f)$ can be well-described by the quantities $\mu'_{rot,sd}(f)$ and $\mu''_{rot,sd}(f)$, as predicted through Equation (29). We can thus immediately calculate the corresponding energy loss value $W_{rot,sd}(f)$ using Equation (13). The results are shown to predict the high-frequency behavior of the energy loss, as illustrated for the 3000 ppm CoO-enriched 1.30 m thick Mn-Zn sample in Figure 14b. The eddy current loss is negligible in such thin samples, but there are no intrinsic limitations regarding thicker samples, because one can easily calculate the quantity $W_{cl,eddy}(f)$ using Equation (10) and subtract it from the measured $W(f)$. Such a simple solution applies, for example, to the high-frequency portion of the $W(f)$ curve in Figure 14b pertaining to

the 4.97 mm thick sample. We can finally obtain, as shown in Figure 14b, the excess loss $W_{\text{exc,sd}}(f) = W(f) - W_{\text{h}} - W_{\text{rot,sd}}(f)$.

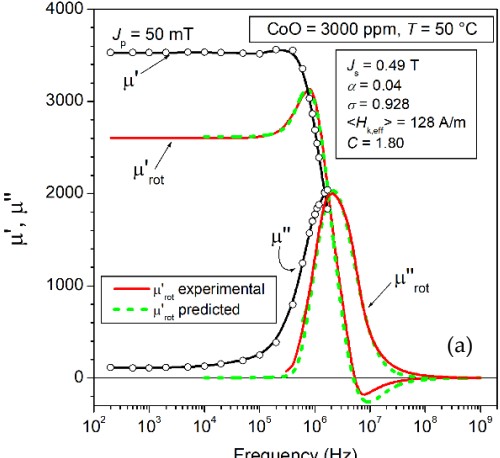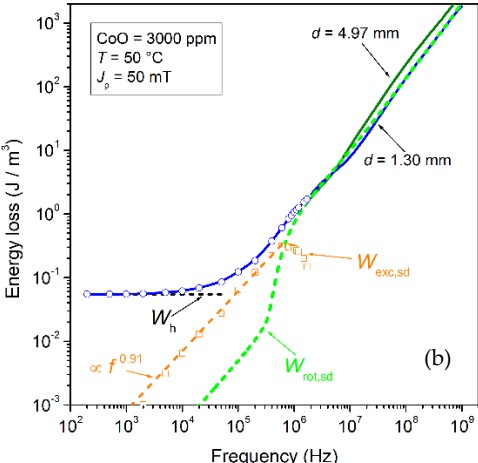

**Figure 14.** Decomposition of permeability and energy loss in a 4000 ppm CoO Mn-Zn ferrite. (**a**) The rotational permeability components $\mu'_{\text{rot}}(f)$ and $\mu''_{\text{rot}}(f)$ are extracted from the measured $\mu'(f)$ and $\mu''(f)$, according to the method discussed in Section 2.2. They are fully described in terms of $\mu'_{\text{rot,sd}}(f)$ and $\mu''_{\text{rot,sd}}(f)$, according to Equation (29). (**b**) The energy loss component $W_{\text{rot,sd}}(f)$, correspondingly calculated using Equation (13), equally fits the high-frequency losses in the thin ($d = 1.30$ mm) eddy-current free samples. The excess loss is finally obtained by subtraction: $W_{\text{exc,sd}}(f) = W(f) - W_{\text{h}} - W_{\text{rot,sd}}(f)$. The power-law dependence on frequency $W_{\text{exc,sd}}(f) \propto f^{0.91}$ is predicted by Equation (32), with the parameter $m = 0.1$.

This quantity exhibits a power-law dependence on frequency $W_{\text{exc,sd}}(f) \propto f^{0.91}$ up to the point where, on entering the MHz range, the dw processes fully relax and both $W_{\text{exc,sd}}(f)$ and $W_{\text{h}}$ drop to negligible values. As far as the $W_{\text{exc,sd}}(f)$ follows the regular power law, Equation (32) applies. The identification of the loss components makes it clear that the monotonical decrease in $W(f)$ versus $T$ observed in all materials down to a minimum at 85 °C (see Figure 10b) is due to the decrease in $W_{\text{h}}$ and $W_{\text{exc,sd}}(f)$. We provide in Figure 15a an example of a loss decrease following the increase in temperature from 0 °C to 85 °C. Such a decrease, limited to frequencies lower than 10 MHz, occurs in spite of a correspondingly predicted increase in $W_{\text{rot,sd}}(f)$ at intermediate frequencies (100 kHz–2 MHz), which is more than compensated by the lowering of $W_{\text{h}}$ and $W_{\text{exc,sd}}(f)$. Such a rule equally applies if the magnetic softening is obtained by an optimal CoO addition (Figure 12). The different behaviors of the dw and rotational losses are related to their different dissipation mechanisms. The dws are affected by a frictional torque, as given by Equation (25), proportional to the wall velocity and, in the absence of inertial effects, they display Debye-type relaxation upon increasing frequency, with a cutoff depending on the ratio between the restoring and damping constants. With the damping constant $\beta_{\text{sd}}$, proportional to the dw energy, any decrease in the anisotropy constant leads, for the same $J_{\text{p}}$ value, to a decrease in the dw-related loss. The precessional motion of the spins inside the magnetic domains is characterized, depending on the local value of $H_{\text{k,eff}}$, by resonant absorption of energy at a frequency $f_0 = \left(\frac{1}{2\pi}\right)\gamma\mu_0 H_{\text{k,eff}}$. The distribution of the $H_{\text{k,eff}}$ values and the related damping effect, embodied by the constant $\alpha_{\text{LL}}$, combine to provide a broadened resonance spectrum, where the $W_{\text{rot,sd}}(f)$ behavior is characterized by a relatively sharp increase at intermediate frequencies, in coincidence with the approach to the peak value of the lognormal distribution function $g(H_{\text{k,eff}})$. It turns out that, by moving $H_{\text{k,eff}}$ toward lower values through magnetic softening treatments, the resonance spectrum is correspondingly enriched at lower frequencies, together with the associated absorption of energy. The related downward trend of $W_{\text{exc,sd}}(f)$ and $W_{\text{h}}$ with temperature, compensating

for the increase in $W_{\text{rot,sd}}(f)$, can be observed in the example shown in Figure 16. The decrease in the anisotropy energy under increasing $T$ (see Figure 12a) is the main physical property lying behind the corresponding decrease in $W_{\text{exc,sd}}(f)$ and $W_{\text{h}}$ [49]. A correlation between the behaviors of these two dw-associated loss components is apparent in Figure 16. This is indeed a general conclusion of the statistical theory of losses, making the excess loss dependent on the distribution and strength of the pinning fields [50]. The statistical parameter $V_0$ in Equation (32) lumps such a dependence, which is related to $W_{\text{h}}$ through the parameter $a(J_{\text{p}})$ in Equation (7).

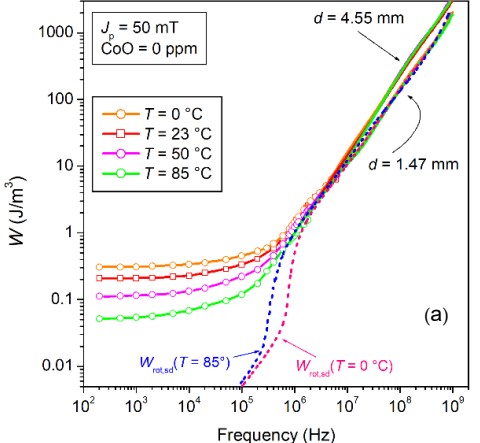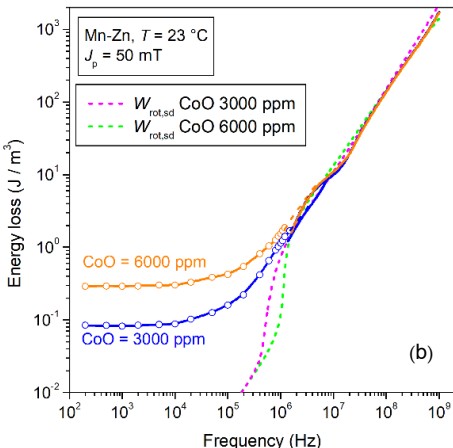

**Figure 15.** The rotational loss component $W_{\text{rot,sd}}(f)$, calculated with Equation (29), predicts the high-frequency magnetic losses in thin samples and thick ones, once the latter are deprived of the eddy current loss contribution. The theory of $W_{\text{rot,sd}}(f)$ predicts, however, that the magnetic softening induced by a distribution of the effective anisotropy widening towards low values is conducive to higher rotational losses at intermediate frequencies (usually between some 100 kHz and a few MHz), which partially compensate for the companion decrease in the dw-generated contributions $W_{\text{h}}$ and $W_{\text{exc,sd}}(f)$. We then observe that the decrease in $<K_{\text{eff}}>$, be it induced by an increase in temperature (**a**) or by optimal CoO addition (**b**), leads to opposite dissipative responses by rotations and dw displacements.

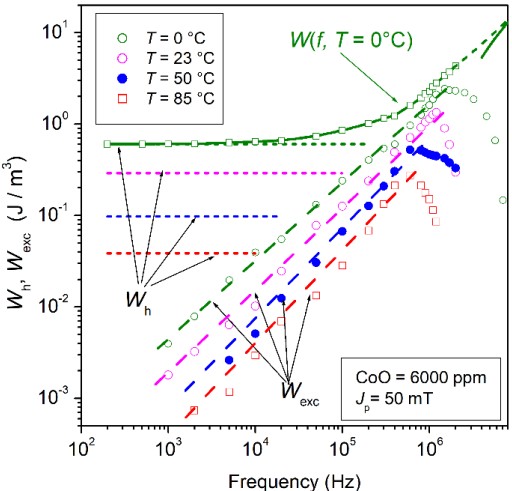

**Figure 16.** Hysteresis $W_{\text{h}}$ and excess $W_{\text{exc,sd}}(f)$ loss components versus temperature ($J_{\text{p}} = 50$ mT, $T = 0, 23, 50, 85\ ^{\circ}$C) in a 6000 ppm CoO-doped Mn-Zn ferrite. The decrease in the effective anisotropy with increasing $T$ justifies the associated decrease in the dw-related loss contributions. The correlation between the $W_{\text{h}}$ and $W_{\text{exc,sd}}(f)$ dependences on $T$ confirms that both quasi-static and dynamic dw displacements respond the same way to strength and distribution of the pinning fields. (Partly adapted from Ref. [42]).

### 3.3. Magnetic Losses in Ni-Zn Ferrites

The previously introduced Figures 1 and 2 bring to light the different frequency responses of the Mn-Zn and Ni-Zn ferrites. The latter display an order of magnitude increase in the cutoff frequency with respect to the former, a property coming at the expense of a large decrease in the initial DC permeability in substantial agreement with Snoek's rule. Where working frequencies around or larger than 10 MHz are required in applications, Ni-Zn ferrites are the material of choice also in view of the absence of any eddy current losses. The conductivity attains in fact negligible levels ($\sigma_{DC} \sim 10^{-5}$ $\Omega^{-1}$ $m^{-1}$ in the discussed commercial EPCOS 4C65 samples at room temperature vs. $\sigma_{DC} \sim 1$–0.1 $\Omega^{-1}$ $m^{-1}$ in typical Mn-Zn ferrites). Notably, the grain size of the order of 1–2 $\mu$m is about one order of magnitude lower than in Mn-Zn ferrites. This points to an increased role of the internal magnetic discontinuities and the related demagnetizing fields in addition to a largely increased coercive field (about 200–300 A/m vs. 5–15 A/m). The anisotropy energy plays an obvious role in coercivity as well as in the general evolution of the energy loss versus frequency by interfering with the balance between rotations and dw displacements. The comparison between the broadband loss behaviors of the 4C65 Ni-Zn and the N87 Mn-Zn ferrites in Figure 1b illustrates the peculiar responses of the Ni-Zn samples. We summarize the main points: (1) $W(f)$ is more than one order of magnitude larger in 4C65 at low frequencies and does not reach the limiting lower value $W_h$ even for $f$ as low as 0.5 Hz; (2) $W(f)$ decreases with frequency up to about 1 MHz, to become persistently lower than the loss of the Mn-Zn ferrite; (3) The FMR resonance spectrum in 4C65 moves toward large frequencies. $\mu'(f)$ peaks now around 15 MHz (see Figure 17a), compared to a few hundred kHz in Mn-Zn. It is then understood, as previously stressed (Figure 15), that the rotational loss $W_{rot,sd}(f)$ retreats towards high frequencies above about 10 MHz, giving way to the dw contribution at intermediate frequencies. This is substantiated by the predicted $W_{rot,sd}(f)$ behavior at room temperature (Equation (29) shown in Figure 18b; (4) The Ni-Zn ferrite is affected by an additional loss term $W_{af}(f)$ generated by the dws. Figure 17a shows a relaxation effect at low frequencies, identifying a loss mechanism with an average time constant of the order of 0.1 s at room temperature. This mechanism, the diffusion after-effect, is temperature-dependent, as illustrated in Figure 18, with activation energy $E_a$ = 0.66 eV; and (5) A further, relatively minor loss contribution appears at low $J_p$ values below about 10 mT, which peaks, as shown in Figure 17b, between some hundred kHz and 1 MHz. The center frequency is relatively independent of temperature (Figure 18b) while decreasing with increasing $J_p$. One might tentatively attribute this contribution to the absorption of energy by dw resonance.

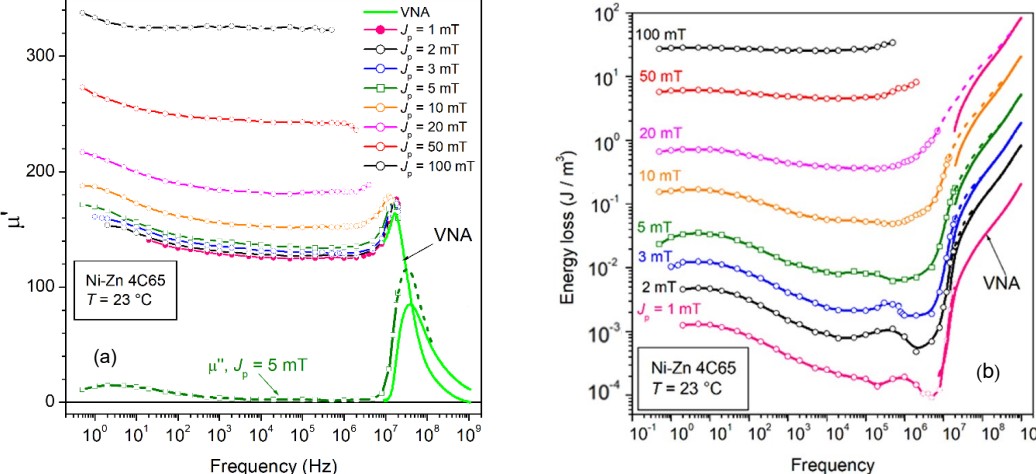

**Figure 17.** (**a**) The frequency dependence of $\mu'(f)$ in the Ni-Zn ferrite 4C65 demonstrates the existence of a relaxation effect, whose average time constant is $\tau \sim 0.1$ s. It is a thermally assisted process of diffusion, which interferes with the motion of the dws and gives rise to relevant extra

losses, as shown in (**b**). A maximum loss value is attained at a frequency $f_0 = 1/2\pi\tau$. A further loss contribution, peaking between 100 kHz and 1 MHz is observed at $J_p$ values lower than 10 mT. This could derive from dw resonance and the related energy absorption. The dashed lines in these figures are reasonable interpolations, which cover those high-frequency intervals that cannot be tested by fluxmetric methods and, having residual dw contributions, do not coincide with the VNA results (solid lines).

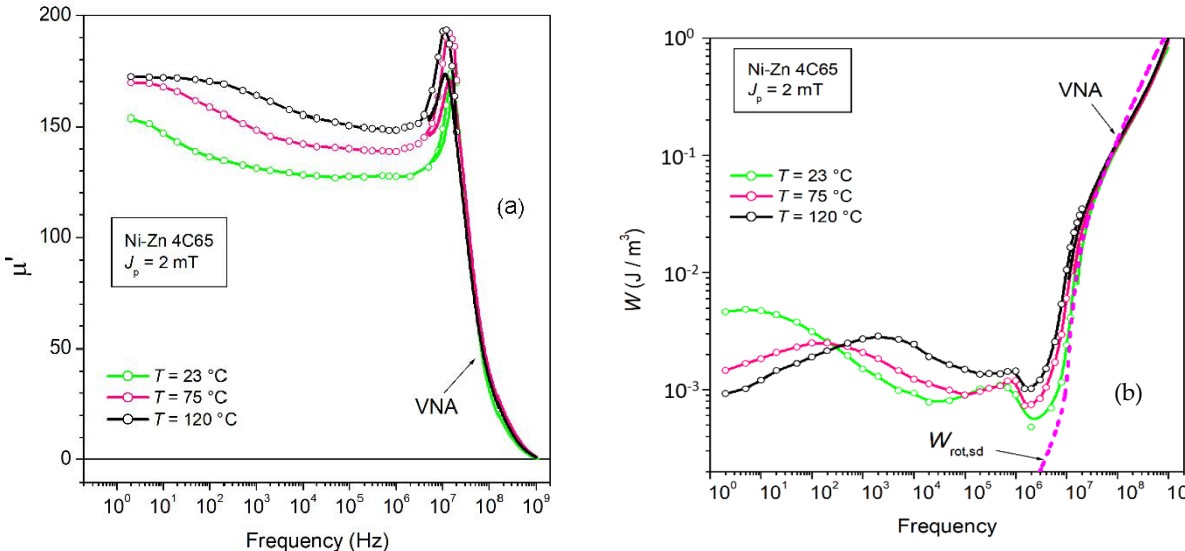

**Figure 18.** The dw relaxation effect associated with the interaction of changing magnetization with diffusing atomic species (diffusion aftereffect) is found to be associated with an activation energy of 0.66 eV. We accordingly observe the cutoff frequency of $\mu'(f)$ (**a**) and the related low-frequency peak of $W(f)$ (**b**) to drift toward higher frequencies upon increasing temperatures. A secondary peak, appearing at all temperatures around 800 kHz, could be related to dw resonance. The predicted rotational loss $W_{\mathrm{rot,sd}}$ (Equation (29)) shows that the resonant absorption of energy by the precessing spins in the bulk drifts towards 10 MHz and beyond.

## 4. Conclusions

We have reviewed the basic phenomenology of magnetic losses in soft ferrites and its theoretical assessment across the broad frequency range DC—1 GHz. Emphasis has been placed on the role of the involved magnetization processes, dw motion and spin rotations inside the magnetic domains, and the related dissipation mechanisms. These are identified in Mn-Zn ferrites with spin damping and eddy currents, with the latter depending on sample thickness and providing a contribution only in the upper-frequency range. The Ni-Zn ferrites are not affected by eddy currents, but fast cation diffusion becomes the source of important viscous hindering of the dw motion and large loss contribution at low-medium frequencies. Whatever the case, interpretation and analysis of the measured loss $W(f)$ rely on the concept of loss separation written in the most general form as $W(f) = W_{\mathrm{h}} + W_{\mathrm{eddy}}(f) + W_{\mathrm{r}}(f)$.

Besides providing a remarkable simplification for the theoretical assessment of $W(f)$, this equation is shown to descend from the very character of the physical process of magnetization. The quasi-static loss $W_{\mathrm{h}}$, exclusively associated with the dw displacements, is easily separated in Mn-Zn ferrites by extrapolating $W(f)$ to $f = 0$ and can be justified by resorting to simple modeling (Equation (7). The eddy current losses $W_{\mathrm{eddy}}(f)$ are of demonstrably classical nature, that is, pertaining to the macroscopic eddy current patterns, and cover, depending on sample thickness, the uppermost portion of the investigated frequency range. They can be analytically predicted and, once calculated, we are left with the so-called residual loss $W_{\mathrm{r}}(f)$. This results, in turn, in the combination of the resonant ab-

sorption of energy by the precessing spins in the bulk and the frictional torque affecting the dynamics of the magnetic moments inside the moving walls $W_r(f) = W_{rot,sd}(f) + W_{exc,sd}(f)$. The rotational term $W_{rot,sd}(f)$ can be calculated starting from the determination of the real and imaginary permeability components as solutions of the Landau–Lifshitz equation and their averaging upon the angular and modulus distributions of the effective anisotropy field $H_{k,eff}$. These solutions are substantiated by the independent experimental separation of the rotational ($\mu'_{rot}(f)$, $\mu''_{rot}(f)$) and dw ($\mu'_{dw}(f)$, $\mu''_{dw}(f)$) permeabilities. The dw-related contribution $W_{exc,sd}(f) = W_r(f) - W_{rot,sd}(f)$ depends on frequency as $W_{exc,sd}(J_p, f) \propto f^n$, with $n \sim 0.9$, and can be given an approximate analytical formulation derived from the statistical theory of losses. This analysis partially applies to the Ni-Zn ferrites because of the overwhelming role of the loss contribution $W_{af}(f)$ by the diffusion aftereffect. It is remarkably predicted, however, that the large increase in $<H_{k,eff}>$ (more than one order of magnitude) with respect to the Mn-Zn ferrites is eventually conducive to lower losses in the MHz range.

Applicative examples of loss modeling drawing from the previous concepts and methods are provided regarding the effect on permeability and losses of the sintering process and the working temperature. It is shown that the assessment of the loss properties passes through the prediction of the effective anisotropy, which is quantitatively derived and shown to exhibit a softened temperature dependence upon the addition of CoO in suitable proportions.

We can then say, in conclusion, that Mn-Zn ferrites with suitable CoO doping (3000–4000 ppm) appear to be the most promising materials, both in terms of the absolute values of energy loss and permeability and because of their moderate sensitivity to the temperature. These properties make them competitive, regarding the broadband magnetic behavior, with amorphous and nanocrystalline cores.

Additional results (Figures S1–S12, Tables S1 and S2) are provided as Supplementary Material.

**Supplementary Materials:** The following supporting information can be downloaded at: https://www.mdpi.com/article/10.3390/magnetochemistry8060060/s1, Figure S1: SEM micrographs of the grain structure of the Mn-Zn ferrites following different sintering treatments: (A) 1325 °C-3 h, (D) 1340 °C-3 h; (G) 1360 °C-3 h; (I)1360 °C-7 h. Figure S2: Distribution of the area fraction of the grains obtained by SEM analysis. Figure S3: (a) Real $\rho'(f)$ and imaginary $\rho''(f)$ resistivity components measured between DC and 10 MHz in Mn-Zn ferrites endowed with different CoO content. The results point to a capacitive components originating from the dielectric properties of the grain boundaries. (b) The DC resistivity increases monotonically with the CoO content. Figure S4: Magnetic characterization of ferrite rings by the fluxmetric method. Figure S5: Measurement of the scattering parameter $S_{11}$ and its dependence on the complex permeability of the ferrite ring sample by a Vector Network Analyzer. The sample is placed at the bottom of a shorted line. The complex permeability is obtained from the sample impedance $Z = \frac{1+S_{11}}{1-S_{11}} Z_0$, where $Z_0$ is the characteristic impedance of the line. Figure S6: The properties of ferrite ring samples sintered at 1325 °C and 1360 °C are compared in this figures. (a) Energy loss up to 1 GHz for Jp ranging between 2 mT and 200 mT. (b) The real $\mu'(f)$ and imaginary $\mu''(f)$ initial permeabilities decrease upon sintering at the higher temperature of 1360 °C. Figure S7: Energy loss $W(f)$ versus frequency in an undoped and an optimally doped Mn-Zn ferrites. The symbols indicate the results of the fluxmetric measurements. The continuous lines, obtained by the transmission line measurements, cover the upper frequency band, till 1 GHz. The overlapping region, belonging to both methiods, is apparent. Eddy currents can provide an extra-contribution to the loss beyond a few MHz in sufficiently thick samples, as signalled by the bifurcating behaviour of $W(f)$ at high frequencies. Figure S8: The rotational contributions $\mu'_{rot}(f)$ and $\mu''_{rot}(f)$ are singled out from the measured permeability in CoO-doped Mn-Zn ferrites. These quantities are theoretically predicted starting from the Landau-Lifshitz equation, which is applied under the assumption that the polycrystalline material is the seat of internal anisotropy fields distributed in direction and amplitude. Figure S9: Energy loss $W(f)$ at $J_p = 50$ mT at different temperatures and different doping levels. Bifurcation at high frequencies separates the response of the as-obtained samples (thickness~5 mm) from that of the same samples after thinning (thickness~1.4 mm). Figure S10: The optimally CoO-doped Mn-Zn ferrites (CoO = 3000–4000 ppm) display the weakest dependence of the

initial permeability on temperature. Figure S11: Examples of loss decomposition in samples with different CoO contents. The hysteresis loss $W_h$ coincides with the limit of the measured loss $W(f)$ for $f \to 0$. The rotational loss $W_{rot}(f)$ is theoretically calculated by means of Equations (29) and (30). The dw generated excess loss $W_{exc}(f)$ is obtained by subtracting $W_h$ and $W_{rot}(f)$ from $W(f)$. It is analytically expressed by Equation (32) (dashed line). Figure S12: Loss decomposition in samples with different CoO contents. The hysteresis loss $W_h$ coincides with the limit of the measured loss $W(f)$ for $f \to 0$. The rotational loss $W_{rot}(f)$ is calculated with Equation (2), using the real and imaginary permeability components. The dw generated excess loss $W_{exc}(f)$ is obtained by subtracting $W_h$ and $W_{rot}(f)$ from $W(f)$. It is analytically expressed by Equation (32); Table S1: Physical parameters of the investigated CoO-doped Mn-Zn ferrites. Table S2: Selected magnetic parameters in the Mn-Zn ferrites versus CoO content at room temperature. $J_s \equiv$ saturation polarization; $\mu_{r,2mT} \equiv$ initial permeability at 100 kHz; $\mu'_{r,100mT} \equiv$ real permeability at 100 mT and 100 kHz; $W_{100mT} \equiv$ Energy loss at 100 mT and 100 kHz; $\mu_{DC,rot} \equiv$ DC rotational permeability; $K \equiv$ calculated average anisotropy constant

**Author Contributions:** Conceptualization, F.F.; Methodology, S.D., C.B., V.T. and F.F.; Software, F.F. and S.D.; Validation, S.D., C.B. and F.F.; Formal Analysis, F.F. and S.D.; Investigation, C.B., S.D. and V.T.; Resources, C.B. and V.T.; Data Curation, S.D., F.F. and V.T.; Writing—Original Draft Preparation, F.F.; Writing—Review and Editing, S.D., C.B., V.T. and F.F.; Supervision, F.F.; Project Administration, C.B.; Funding Acquisition, C.B., F.F. and S.D. All authors have read and agreed to the published version of the manuscript.

**Funding:** This research work was partially supported by the 19ENG06 HEFMAG project, which was funded by the EMPIR program, and co-financed by the Participating States and the European Union's Horizon 2020 research and innovation program.

**Institutional Review Board Statement:** Not applicable.

**Informed Consent Statement:** Not applicable.

**Data Availability Statement:** Representative data, in excess of the data reported in the article, are here available as Supplementary Material.

**Acknowledgments:** This work was performed in the context of the Agreement of Scientific and Technological Cooperation between the Istituto Nazionale di Ricerca Metrologica (INRIM) and The Centre for Research and Technology Hellas (CERTH) regarding the study of soft ferrites. S. Dobák acknowledges the support of stay at the INRIM laboratories from The National Scholarship Program of the Slovak Republic.

**Conflicts of Interest:** There are no conflict of interest.

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
