# Peer review of "Magnetic Losses in Soft Ferrites"

_magnetochemistry, doi:10.3390/magnetochemistry8060060_

Round 1

Reviewer 1 Report

Strengths

The authors have done review the basic phenomenology of magnetic losses from DC to 1 GHz in commercial and laboratory prepared soft ferrites, in the light of recent concepts regarding their physical interpretation.This is based, on the one hand, on the identification of the contributions to the magnetization process provided by spin rotations and domain walls, and, on the other hand, on the concept of loss separation. It additionally contemplates a distinction between the involved microscopic dissipation mechanisms: spin damping and eddy currents. Selected experimental results on the broad band behavior of complex permeability and losses in Mn-Zn ferrites provide significant examples of their dependence on sintering methods, additional solute elements, and working temperature. The physical modeling of the losses brings to light the role of the magnetic anisotropy and the way its magnitude distribution, affected by the internal demagnetizing fields, acts upon the magnetization process and its dependence on temperature and frequency. It is shown that the effective anisotropy governs the interplay of domain wall and rotational processes and their distinctive dissipation mechanisms, whose contributions are recognized in terms of different loss components. Applicative examples of loss modeling drawing from the previous concepts and methods are provided regarding the effect on permeability and losses of the sintering process and of the measuring temperature. It is shown that the assessment of the loss properties passes through the prediction of the effective anisotropy, which is quantitatively derived and shown to exhibit a softened temperature dependence upon addition of CoO in suitable proportions.

Weakness 

Unfortunately, there are unfortunate errors and typographical errors.

  1. Citation: 2022 not 2021, (line 22) and in other pages.
  2. Links to Fig. 6, 16 and Table. 1 after they appear in the text.
  3. An unfortunate expression is magnetic viscosity, (line 17).
  4. Update references.
  5. The conclusions should be more specific. It is necessary to give comparative technical characteristics of the best ferrites from Mn-Zn and Ni-Zn, prepared according to optimal modes. It turned out to be a very deep and extensive theoretical overview without recommendations for the best materials for the consumer.

Reviewer 2 Report

1- There is a massive misunderstanding from my side as a reviewer if this article is a review article or a regular article for the authors?

2- The authors mentioned in the supplementary file attached with the article that they prepared Mn-Zn ferrite. After that, the authors made doping with Nb2O5 and CaO and made different characterization and measurements, so I think this information in the supplementary file should be added to the article.

3- In the article, the authors mentioned other materials (not prepared by the authors), and unclear from where the authors got these samples, and I cannot understand if the authors compare their work with other works or what?

4- I cannot find any reference from where the authors got these results or/and the figures mentioned in the article. If the article is a review article, the authors should mention the used references after each figure title, even if the used references are mentioned before in the text.

5- The authors mentioned more than 30 equations in the article, and they did not provide any direct reference for each equation, which should be noted before or after the equation directly.

Reviewer 3 Report

Overall a very thorough and interesting review of magnetism in soft ferrites. I have comments relating only to clarity and grammar, as follows.

Fig. 2a: label mu' and mu''

line 182: unclear grammar; suggest: "Consequently, at higher Jp values there is a reduction in the frequency overlap interval (see Fig...."

line 347: remove "To" ; "Note that the frequency..."

lines 417-564: line spacing was larger

Fig 6a: explain legends "exp. flxm." and "exp VNA" in caption

line 539: remove "to"; ", where by subtracting the measured..."

line 573: unclear language, rephrase to "...demagnetizing fields, is seen to be fundamental in ..."

caption to figure 8: remove comma on line 4; "structural inhomogeneities, besides moderate..."

line 734: "is the seat of" is unclear here; perhaps "The Ni-Zn ferrite presents an additional loss..."

line 735, and Fig 17 caption: "puts in evidence" also awkward language; suggest "demonstrates" in place of it.

Round 2

Reviewer 2 Report

I am satisfied with the authors' responses to my previous comments and their clarifications in answering the comments. I believe that the article can be published in its present form.